# Long-read direct RNA sequencing reveals epigenetic regulation of chimeric gene-transposon transcripts in *Arabidopsis thaliana*

Jérémy Berthelier [1] ✉, Leonardo Furci [1], Shuta Asai [2], Munissa Sadykova [1], Tomoe Shimazaki[1], Ken Shirasu [2] & Hidetoshi Saze [1] ✉

Transposable elements (TEs) are accumulated in both intergenic and intragenic regions in plant genomes. Intragenic TEs often act as regulatory elements of associated genes and are also co-transcribed with genes, generating chimeric TE-gene transcripts. Despite the potential impact on mRNA regulation and gene function, the prevalence and transcriptional regulation of TE-gene transcripts are poorly understood. By long-read direct RNA sequencing and a dedicated bioinformatics pipeline, ParasiTE, we investigated the transcription and RNA processing of TE-gene transcripts in *Arabidopsis thaliana*. We identified a global production of TE-gene transcripts in thousands of *A. thaliana* gene loci, with TE sequences often being associated with alternative transcription start sites or transcription termination sites. The epigenetic state of intragenic TEs affects RNAPII elongation and usage of alternative poly(A) signals within TE sequences, regulating alternative TE-gene isoform production. Co-transcription and inclusion of TE-derived sequences into gene transcripts impact regulation of RNA stability and environmental responses of some loci. Our study provides insights into TE-gene interactions that contributes to mRNA regulation, transcriptome diversity, and environmental responses in plants.

Transposable elements (TEs) are major components of eukaryotic genomes, influencing the organization of chromosome structure and expansion of genome size[1,2]. TEs can insert into intragenic regions of gene units, including promoters, 5'/3'- untranslated regions (UTRs), exons, and introns, impacting both genetic and epigenetic regulation of gene transcription and transcriptome diversity[3]. For example, TEs can act as regulatory elements of nearby genes in response to environmental stresses[4–6]. Co-transcription and exonization of TE sequences generate chimeric TE-gene (TE-G) transcripts, often resulting in the co-option of TE sequences for the innovation of novel gene products[7].

In the human genome, it has been reported that 4% of protein-coding genes possess TE-derived exon sequences and that 5% of alternatively spliced internal exons are derived from short nuclear-interspersed element *Alu*[8–10]. In contrast, it has been estimated that in *Arabidopsis*, TE sequences are associated with 7.8% of expressed genes, and 1.2% may contribute to the protein-coding regions[11].

Intragenic TE sequences can profoundly impact pre- and post-transcriptional regulation of mRNA[12]. TE sequences affect various alternative splicing (AS) processes of precursor mRNAs, resulting in intron retention (IR), exon skipping (ES), or the creation of alternative

[1]Plant Epigenetics Unit, Okinawa Institute of Science and Technology Graduate University (OIST), 1919-1 Tancha, Onna-son, Kunigami-gun, Okinawa 904-0495, Japan. [2]Center for Sustainable Resource Science, RIKEN, 1-7-22 Suehiro-cho, Tsurumi, Yokohama, Kanagawa 230-0045, Japan. ✉e-mail: berthelier.j@laposte.net; hidetoshi.saze@oist.jp

splice donor (or alternative 5′ splicing site; A5SS) and alternative splice acceptor sites (or alternative 3′ splicing site; A3SS)[13]. In addition, the internal promoter of a TE sequence can act as an alternative transcription start site (ATSS), while TE sequences in introns or 3′-UTRs can create alternative transcription termination sites (ATTS) owing to the presence of alternative polyadenylation (APA) signals within TE sequences[14]. In *Arabidopsis*, ~40% of intron-containing genes are alternatively spliced[15], and tissue-specific and stress-responsive IR and ES events have been detected in a subset of loci[16].

To suppress the harmful effects of active TEs, plants and other organisms have evolved epigenetic mechanisms, such as DNA methylation and histone modifications[17]. In plants, DNA is methylated at cytosines in CG and non-CG contexts (CHG and CHH, where H can be A, T, C or G). CG methylation is maintained by METHYLTRANSFERASE 1 (MET1), while non-CG methylation is in part regulated by the histone H3K9 methylase KRYPTONITE/SUVH4 together with SUVH5 and SUVH6[17]. The chromatin remodeler DECREASE IN DNA METHYLATION 1 (DDM1) is required for the maintenance of both DNA methylation and H3K9 methylation[18]. On the other hand, ectopic accumulation of H3K9 methylation and non-CG methylation in genic regions are prevented by the H3K9 demethylase INCREASE IN BONSAI METHYLATION 1 (IBM1)[19].

In mammals, changes in DNA methylation and histone modification induce spurious transcription initiation from ATSSs in TE sequences[20,21]. In addition, a range of cancer cell types overexpress TE-derived alternative splice variants of oncogenes[22]. In plants, several transcription start sites (TSSs) of maize genes have been identified within TEs[6], and in *Arabidopsis thaliana*, epigenetically regulated cryptic TSSs are mostly embedded in TE sequences[23]. DNA methylation of intronic TEs has been shown to affect the mRNA splicing of genes that regulate the seed coat of soybean and the alternative poly(A) sites required for fruit development of oil palm[24,25]. Previous studies have shown that intronic TEs tend to be accumulated in plant disease resistance (*R*) genes, associated with repressive chromatin marks, including DNA methylation and H3K9 methylation, in the *Arabidopsis* and rice genomes[26,27]. These heterochromatic intragenic TEs are regulated by a protein complex that comprises ENHANCED DOWNY MILDEW 2 (EDM2), INCREASE IN BONSAI METHYLATION 2 (IBM2/ASI/SGI), and ASI1-IMMUNOPRECIPITATED PROTEIN 1 (AIPP1)[27–34]. On the other hand, TEs inserted in the 3′-UTR regions of genes often regulate mRNA stability, translation, and subcellular localization[35–38].

Despite the potential importance of TE-G transcripts in plant developmental traits and environmental responses, limitations of short-read RNA sequencing and a lack of bioinformatic pipelines have hindered the detection and comprehensive analyses of these transcripts in plants. The recent development of long-read sequencing technologies now permits in-depth analyses of the complexity of mRNA transcription and processing dynamics[39–44], and transposon regulation[45]. In this study, we employed Oxford Nanopore Direct RNA Sequencing (ONT-DRS) technology to investigate the prevalence of TE-G transcripts in the *A. thaliana* transcriptome. By developing a new bioinformatic tool, ParasiTE, we identified TE-G transcripts associated with AS, ATSS, and ATTS. Additionally, we investigated the epigenetic and environmental regulation of TE-G transcripts using a mutant panel and public transcriptome dataset. Our study revealed a global production of TE-G transcripts generated from about 3000 gene loci in the *Arabidopsis* transcriptome, with many TEs associated with ATSS and ATTS being found in 5′/3′-UTRs. We also demonstrate that TE sequences in 3′-UTRs contribute to the response of genes to environmental signals and regulation of RNA stability.

## Results

### DRS of the *Arabidopsis* transcriptome
To understand the impact of TEs on RNA processing and the prevalence of chimeric TE-G transcript formation in the *Arabidopsis* transcriptome, we exploited ONT-DRS, which allows for native long-transcript sequencing[46]. DRS with wild-type Columbia (Col-0) seedlings (Supplementary Data 1) could capture long chimeric TE-G transcript and alternative TE-G (ATE-G) isoforms, such as *Resistance to Peronospora parasitica 4* (*RPP4*)-*ATCOPIA4* transcripts in wild-type[32,43,47–49], which have not been represented in the precedent annotations of *Arabidopsis* transcript isoforms in Araport11[50] or the latest comprehensive *Arabidopsis* transcriptome dataset, AtRTD3[44] (Fig. 1a, b, Supplementary Fig. 1a). However, the low coverage of the DRS dataset may hinder the comprehensive detection of potential TE-G transcripts. To circumvent the issue, we combined publicly available wild-type Col-0 DRS data[40] with our wild-type Col-0 DRS data to increase read coverage for the detection of TE-G transcripts (Supplementary Table 1). The DRS dataset was further merged with either the Araport11 or AtRTD3 transcript dataset, yielding the unique transcriptome annotation datasets DRS-Araport11 and DRS-AtRTD3 (see Methods for details) (Fig. 1a; Supplementary Fig. 1b; Supplementary Table 2, 3; data available at https://plantepigenetics.oist.jp/). DRS-Araport11 and DRS-AtRTD3 covered about 20% more genes with TE-G transcripts than the original Araport11 and AtRTD3 datasets (Supplementary Table 4). We mainly employed the DRS-AtRTD3 transcriptome dataset for downstream analyses since it showed a higher number of TE-G transcripts than DRS-Araport11 (Supplementary Fig. 1c; Supplementary Table 4; Supplementary Data 2).

### ParasiTE: a tool for the detection of TE-G transcripts and their alternative isoforms
Although there are various bioinformatic tools for identifying alternative promoters of genes provided by TEs[51,52], to our knowledge, no tools existed that had been specifically designed to detect TE-G transcripts and their alternative isoforms from transcriptome datasets according to the associated RNA processing events. Therefore, we developed a new tool named ParasiTE to detect TE-G transcripts and TE-G transcripts with alternative transcript isoforms (ATE-G isoforms; Fig. 1b, c) in transcriptome datasets. ParasiTE further classifies ATE-G isoforms into TE-associated alternative splicing (TE-AS) and TE-associated alternative transcript production (TE-ATP) according to the associated RNA processing events[53] (Fig. 1c). TE-AS events include TE-associated intron retention (TE-IR), exon skipping (TE-ES), the creation of alternative 5′ splicing site (TE-A5SS), and of alternative 3′ splicing site (TE-A3SS). TE-ATP events include TE-associated alternative transcription start site (TE-ATSS), and alternative transcription termination sites (TE-ATTS). ParasiTE further distinguishes the TE-associated alternative first exon (TE-AFE) from TE-ATSS and TE-associated alternative last exon (TE-ALE) from TE-ATTS[53] (Fig. 1c; Supplementary Note 1). Validation of ParasiTE with Araport11 transcript annotation and manual inspection of the output data demonstrated that ParasiTE could detect TE-ATP events with higher accuracy than TE-AS events (Supplementary Note 1).

### Identification and classification of TE-G transcripts in *A. thaliana*
Analysis of TE-G transcripts detected by ParasiTE revealed a total of 11,348 TE-G transcripts and 6,025 ATE-G isoforms in DRS-AtRTD3 (Fig. 1b). We found that about 17% of *A. thaliana* TEs were involved in TE-G transcript formation, and 7% of which were transcribed as ATE-G isoforms (Fig. 1d; Supplementary Fig. 1d; Supplementary Table 2, 4). About 8% of genes were associated with TE-G transcripts, which was similar to a previously reported estimation (7.8%)[11]. Among the TE-G transcripts, we found that 3% of genes were involved in ATE-G isoform production (Fig. 1d; Supplementary Fig. 1d; Supplementary Table 2, 4). The Gene Ontology (GO) analysis of biological functions of the TE-G transcripts suggested an enrichment in terms related to plant defense responses against pathogens, such as Cell killing and Defense response to fungus, while ATE-G isoforms were implicated in Pol II-dependent RNA transcription (Fig. 1e).

ParasiTE can classify ATE-G isoforms according to the associated RNA processing events (Fig. 1c, f, g). Among the TE-AS, TE-IR was the

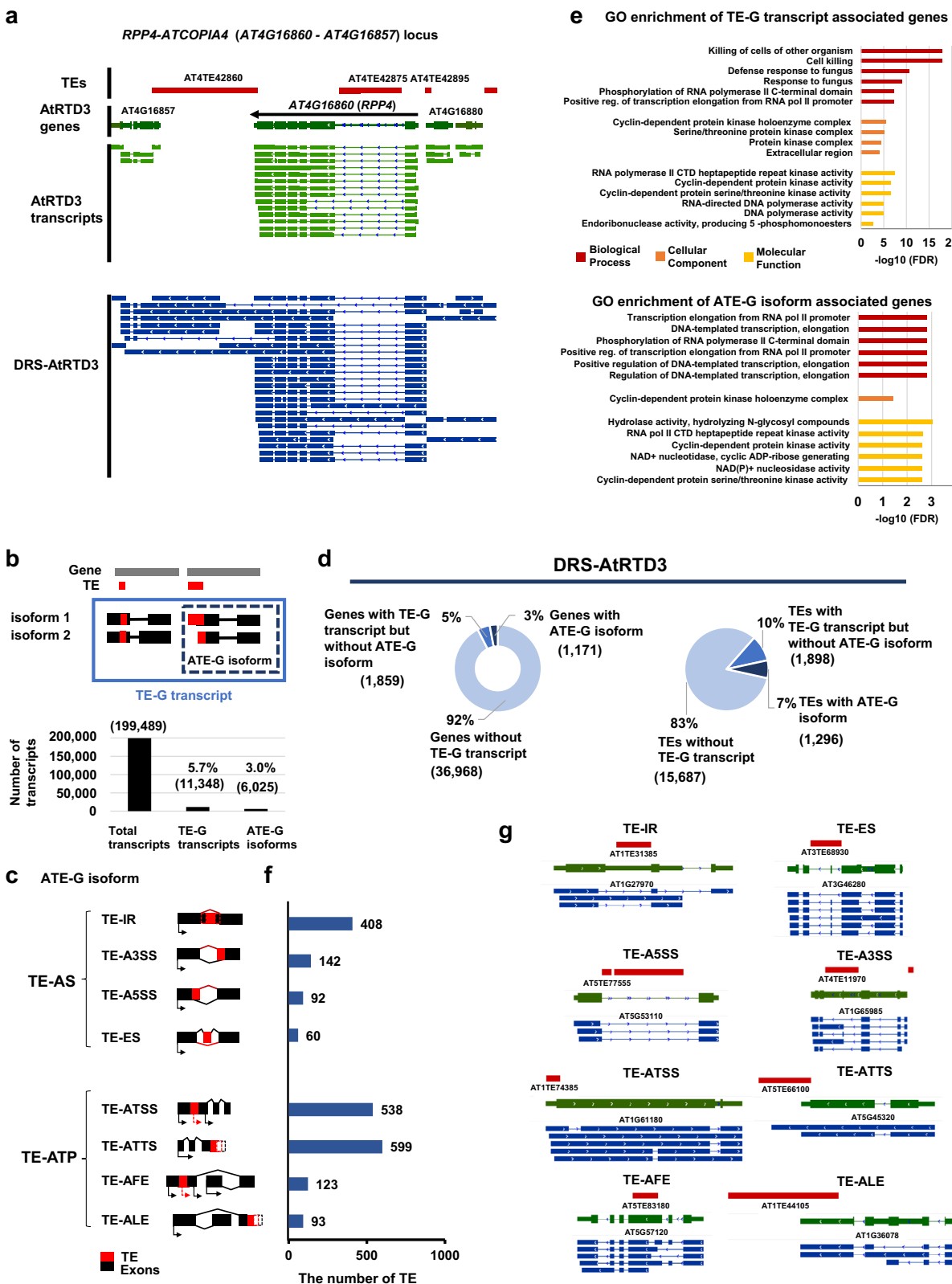

more frequent event (Fig. 1f, g; Supplementary Fig. 1d). Interestingly, we found more TEs involved in TE-ATP events with TE-ATSS and TE-ATTS than TE-AS (Fig. 1f, g; Supplementary Fig. 1d). Indeed, we found that many TE sequences were associated with the first and last exon of transcripts (Supplementary Fig. 2a), overlapping with 5′/3′-UTR of gene transcripts (Supplementary Fig. 2b). These results demonstrate that ParasiTE can efficiently detect chimeric TE-G transcripts and that

TE sequences nearby or within genes contribute as a source for transcriptome diversity in *Arabidopsis*.

**Contribution of TE superfamilies to TE-G transcript production**
ParasiTE detected 3194 exonic and 441 intronic TEs in the DRS-AtRTD3 dataset (Supplementary Fig. 2c). Among them, Helitron and Mutator were abundant families, which were also enriched in TE-G transcript

**Fig. 1 | Detection of TE-gene transcripts (TE-G transcripts) and alternative TE-gene isoforms (ATE-G isoforms) in DRS-AtRTD3 of the *Arabidopsis* transcriptome. a** Representative TE-G transcripts associated with the *RPP4-ATCOPIA4* locus detected in DRS-AtRTD3. **b** The number of TE-G transcripts and ATE-G isoforms detected by ParasiTE. **c** RNA processing events associated with ATE-G isoforms (TE-AS and TE-ATP). TE-AFE and TE-ALE are included in TE-ATSS and TE-ATTS, respectively. AS; Alternative Splicing, ATP; Alternative Transcript Production, IR; Intron Retention, ES; Exon Skipping, A5SS; Alternative 5′ Splicing Sites, A3SS; Alternative 3′ Splicing Sites, ATSS; Alternative Transcription Start Sites, ATTS; Alternative Transcription Termination Sites, AFE; Alternative First Exon, ALE;

Alternative Last Exon. **d** The number of genes (left; 39,998 gene models based on DRS-AtRTD3 annotation; 27,628 genes were associated with Arabidopsis Genome Initiative codes) and TEs (right; *n* = 18,881 TE annotations) associated with TE-G transcripts and ATE-G isoforms identified in DRS-AtRTD3. **e** Enriched Gene Ontology terms of genes associated with TE-G transcripts (top) and ATE-G isoforms in DRS-AtRTD3 (bottom). **f** The number of TEs associated with ATE-G isoform events. Some TEs were included in several ATE-G isoform categories. **g** Representative loci associated with TE-AS and TE-ATP events detected in DRS-AtRTD3. Red, TE annotation; Green, AtRTD3 annotation (collapsed); Blue, DRS-AtRTD3 annotation (extended). Source data are provided as a Source Data file.

and ATE-G isoform events (Fig. 2a; Supplementary Data 3). Generally TE sequences associated with TE-G transcripts and ATE-G isoforms were slightly shorter in average compared with intergenic TEs (Fig. 2b). We then examined DNA methylation levels of intronic TEs, and TEs associated with TE-G transcript and ATE-G isoform by employing a DNA methylation calling method with the Nanopore long-read DNA sequencing data[54], which allows for the detection of inner DNA methylation of long, repetitive TE (Supplementary Fig. 3a, b; Supplementary Data 4). We found a significantly lower DNA methylation especially in TEs associated with TE-AS events compared with intronic TEs (Fig. 2c). These results suggest that hypomethylated TE sequences are preferentially transcribed and incorporated into TE-G transcripts.

TE sequences at exon-intron junctions provided canonical GT-AG sequences as splice donor/acceptor sites (Supplementary Fig. 4). We found an enrichment of DNA transposons such as Mutator, CACTA, and Harbinger superfamilies in TE-AS, including in TE-IR and TE-A5SS events (Fig. 2d, e; Supplementary Fig. 5; Supplementary Data 3). Further analysis of TE families involved in TE-IR showed that they are generally short (<1,000 bp), suggesting that they are degenerated TE sequences retained in gene bodies, providing splicing donor/acceptor sites (Supplementary Fig. 5b). On the other hand, long terminal repeat LTR/Gypsy was under-represented in most of the splicing events, while LTR/Copia was slightly but significantly over-represented, especially in TE-ALE events (Fig. 2d). Overall, these results suggest that TE sequences in intragenic regions can be incorporated into TE-G transcript by providing splicing donor/acceptor sites.

**Alteration of epigenetic modifications impacts TE-G transcript production**

TEs are often activated by changes in epigenetic modifications, which affects the expression of associated genes[23]. To understand the epigenetic regulation of TE-gene transcription, we analyzed ATE-G isoform expression and isoform usage in mutants defective in the regulation of heterochromatic marks, including DNA methylation (*met1* and *ddm1*) and histone H3K9 methylation (*suvh456* and *ibm1*), and also mutants showing defects in the transcription of intragenic TEs (*ibm2* and *edm2*) (Supplementary Fig. 6). By analyzing changes in ATE-G isoform usage in mutants using short-read RNA-seq datasets with the DRS-AtRTD3 reference, we found that approximately 10–20% of genes associated to ATE-G isoforms showed significant differential isoform usage (or isoform switching) in the epigenetic mutants (Fig. 3a; Supplementary Fig. 7). Those ATE-G isoforms are mostly represented by TE-ATSS or TE-ATTS events (Supplementary Fig. 9a, 10a). This number was higher in *met1*, *ddm1*, and *suvh456*, which suggests that DNA methylation and H3K9 methylation regulate the transcription of ATE-G isoforms (Fig. 3a; Supplementary Data 5). We then performed de novo assembly of the DRS data based only on the mutant transcriptomes to identify mutant-specific ATE-G isoforms. DRS with three biological replicates yielded a total of 3.2–5.2 million reads for each mutant (Supplementary Data 1, 6). ATE-G isoforms in the mutant-DRS transcriptome assemblies detected by ParasiTE were further analyzed for differential isoform usage with short-read RNA-seq data, changes in DNA and H3K9 methylation, and association with the IBM2/EDM2 binding peaks obtained by chromatin immunoprecipitation (ChIP)-seq

(Fig. 3a; Supplementary Fig. 6–10 and Supplementary Data 6). As expected, due to the low coverage of the DRS data, a smaller number of ATE-G isoforms was identified in mutant-DRS transcriptome assemblies compared with DRS-AtRTD3 (Fig. 3a; Supplementary Fig. 6a). However, the *ddm1*-DRS transcriptome showed a higher number of ATE-G isoforms than the other mutants, suggesting that epigenetic changes in TEs in this mutant may contribute to mutant-specific TE-G transcript production (Epi-ATE-G isoform). DNA methylation analysis showed that changes in ATE-G isoform usage in *met1* and *ddm1* were associated with a significant reduction of DNA methylation in the TEs (Supplementary Fig. 7–10). We also analyzed changes in DNA methylation in CHG context as a proxy of H3K9 methylation in TEs associated with ATE-G isoforms in *ibm1*, *suvh456*, *ibm2*, and *edm2* (Supplementary Fig. 8–10). While changes in usage of ATE-G isoforms in *suvh456* was associated with a significant reduction of CHG methylation in TEs, TEs with ATE-G isoforms in *ibm1*, *edm2*, and *ibm2* showed no significant changes in CHG methylation (Supplementary Fig. 8–10), consistent with reports that IBM2 and EDM2 act downstream of DNA methylation and H3K9 methylation[27–34]. IBM1 is known to suppress ectopic H3K9 methylation and non-CG methylation in gene body[19,55–57], while it showed limited impact on CHG methylation in TE associated with ATE-G isoform (Supplementary Fig. 8b, c). Overall, these results demonstrated the importance of epigenetic modifications for the regulation of TE-G transcript production.

**Epigenetic regulation of TE-ATSS formation in *A. thaliana***

Cryptic TSSs at intragenic or intergenic regions are epigenetically regulated in *A. thaliana*[23,58,59], many of which are associated with TE sequences[23]. In *ddm1* and in *met1*, we detected a previously reported TE-G transcript in the *SQN* (*AT2G15790*) locus with TE-ATSS events, with activation of *AT2TE28020/25* (TIR/Mutator) in the upstream region inducing various readthrough transcripts with *SQN* (Fig. 3b)[23,60,61]. For *AT2G16050*, similar examples were found, with *AT2TE28420/25/30* (Helitron; Supplementary Fig. 11)[23]. In addition, we identified novel mutant-specific TE-ATSS at several gene loci, including *AT1G75990* with *AT1TE93320* (TIR/Mutator) and *AT3G23080* with *AT3TE34420/30* (TIR/hAT; Supplementary Fig. 11). We also identified a gene encoding single-stranded nucleotide-binding protein RH3 (*AT2G40960*) with *AT2TE77005/15* (LTR/Gypsy; while only 5′ and 3′ LTRs were annotated, we found that the region encodes a complete TE sequence belonging to the Gypsy-39_AT family; Fig. 3b, c). These TEs were hypomethylated in *ddm1* and/or *met1*, and based on the published cap analysis of gene expression (CAGE)-seq[23], the ATE-G isoforms were transcribed from cryptic TSSs. These results demonstrate that epigenetic repression of ATSS associated with intergenic TEs, especially those located in the 5′ region of genes, is essential for the suppression of TE-G transcript production with downstream genes.

**Epigenetic regulation of TE-ATTS formation in *A. thaliana***

Together with TE-ATSS, TE-ATTS events were frequently detected in the transcriptome of DRS-AtRTD3 and of mutant-DRS (Fig. 1c, f). Although TE sequences in promoter regions are known to act as regulatory elements of downstream genes in both animals and plants[2], the impact of TEs inserted in the 3′ region of genes on transcription in

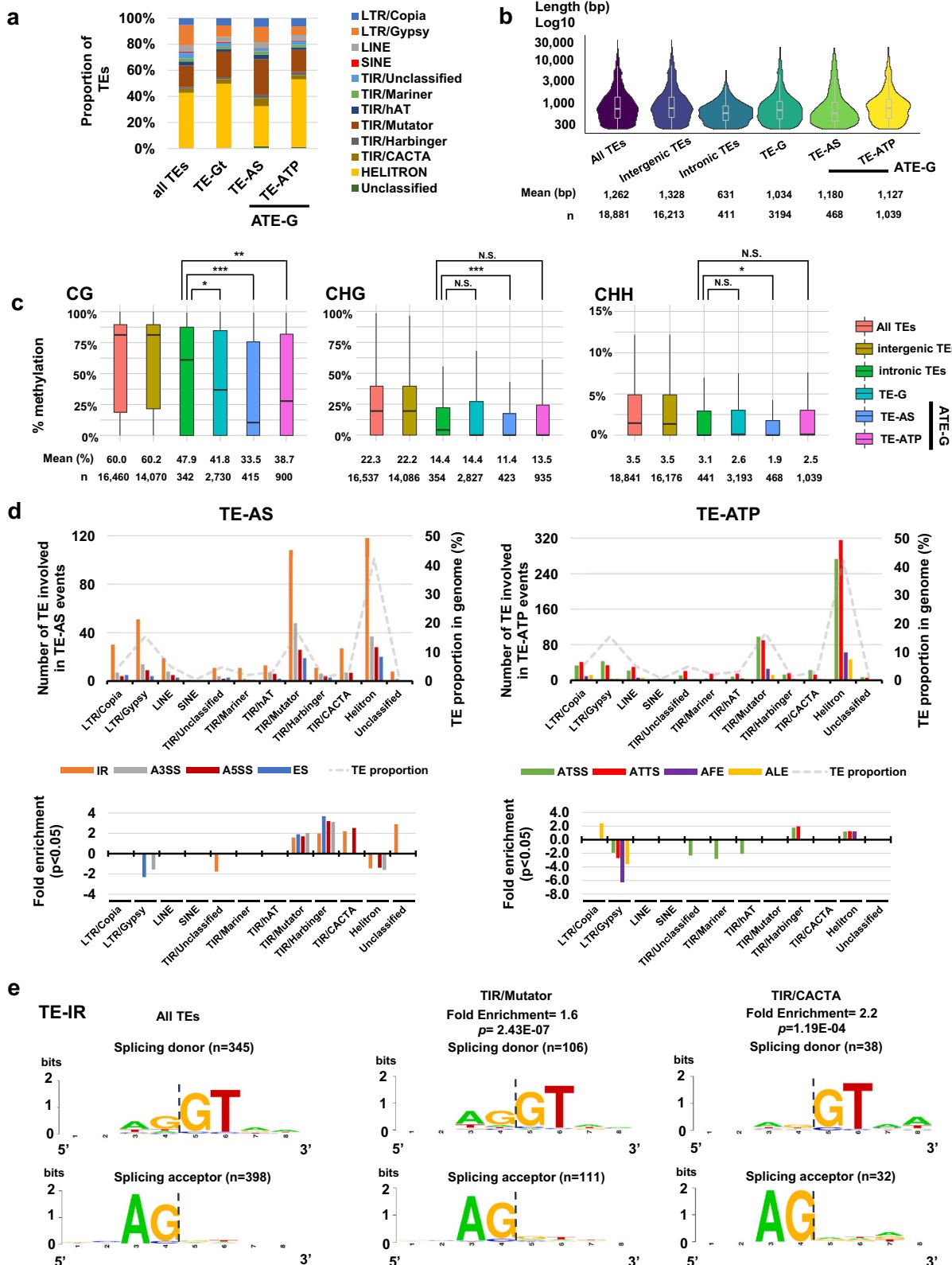

*Arabidopsis* is less well understood. Based on the Col-0 DRS data, we detected poly(A) sites within 420/599 TEs associated with TE-ATTS events (Supplementary Fig. 12a), which may correspond to APA sites of associated genes. Indeed, 437/599 TEs associated with TE-ATTS events overlapped with poly(A) sites in the public APA data for the *Arabidopsis* genome[62] (Supplementary Fig. 12a), suggesting frequent transcription termination at APA in TE sequences.

In the wild-type Col-0, TE-ATTS were formed as a result of read-through transcription and termination at APA(s) in TEs inserted in the 3′ region of genes, including *ATCOPIA4* in *RPP4*, *Gypsy/ATGP2* in *RPP5-like* (*AT4G16900*), and *ONSEN/ATCOPIA78* in both *GEM-RELATED 5* (*GER5; AT5G13200*) and *BESTROPHIN-LIKE PROTEIN1* (*BEST1; AT3G61320*) loci (Fig. 4a; Supplementary Fig. 12–15). Despite active transcription of the TE regions, these TEs were often associated with

**Fig. 2 | TE superfamilies associated with ATE-G isoforms. a** Proportion of TE superfamilies in the *Arabidopsis thaliana* genome, TE-G transcripts, and ATE-G isoforms. **b** Comparison of the length of all TEs, intergenic TEs, intronic TEs, and TEs associated with TE-G transcripts, and with ATE-G isoforms. The centerline represents the median. The borders of the boxplots are the first and third quartiles (Q1 and Q3). Whiskers represent data range, bounded to 1.5 * (Q3-Q1). **c** DNA methylation (CG, CHG, and CHH contexts) of all TEs, intergenic TEs, intronic TEs, and TEs associated with TE-G transcripts, and with ATE-G isoforms. *P*-values were obtained by the Mann-Whitney U test; *, *p* < 0.05; **, *p* < 0.01; ***, *p* < 0.001. The centerline represents the median. The borders of the boxplots are the first and third quartiles (Q1 and Q3). Whiskers represent data range, bounded to 1.5 * (Q3-Q1). **d** Top: the number of TEs associated with ATE-G isoforms (left, TE-AS; right, TE-ATP). The proportion of the TE superfamilies in the genome is also displayed as a reference. Bottom: Fold-enrichment of TE superfamilies significantly enriched in TE-AS or TE-ATP. *P*-values were obtained by the hypergeometric test. Only TE superfamilies with fold-enrichment of *p* < 0.05 are shown. **e** Enrichment of nucleotides at splicing donor and splicing acceptor sites associated with TE-IR for all TEs, and TE superfamilies TIR/Mutator and TIR/CACTA with fold-enrichment of *p* < 0.05. Fold-enrichment and *p* values (hypergeometric test) are indicated. Source data are provided as a Source Data file.

repressive chromatin marks, such as DNA and H3K9 methylation, in the wild-type (Fig. 4a; Supplementary Fig. 9–10, 12c; Supplementary Data 5, 6). TE-ATTS formation was also detected in intronic TEs in the epigenetic mutants, with the loss of heterochromatic marks in intronic TEs resulting in premature polyadenylation of mRNAs at promoter-proximal APAs within TE sequences[26,29,63] of *R* gene *Resistance to Peronospora Parasitica 7* (*RPP7*; *AT1G58602*) and chloroplast protein gene *PPD7* (*AT3G05410*) (Supplementary Fig. 13, 16). In a few loci, however, we detected Epi-ATE-G isoforms showing rather enhanced read-through transcription of 3′ TE sequences in the mutant backgrounds, such as *ibm1*, *edm2*, and *ibm2* (Supplementary Fig. 17, 18).

Our ChIP-seq analysis of RNA Pol II phosphorylated at Ser5 and Ser2 of the carboxy-terminal domain (CTD) repeats showed that in the wild-type Col-0, Pol II can elongate through intronic TEs or TEs in 3′ UTRs, despite the presence of repressive chromatin marks (Fig. 4a, b; Supplementary Fig. 12, 16). *suvh456* caused a loss of Pol II peaks in the TE sequences (Fig. 4a, c; Supplementary Fig. 16), suggesting elongation defects and the release of Pol II from these TE regions. This result suggests that maintenance of heterochromatic modifications such as H3K9 and DNA methylation are required for Pol II elongation through the intragenic TE sequences in these loci. An increase of PolII in TE regions in *met1* (Fig. 4c) might be caused by reactivations of TEs associated with ATTS. Mutants of IBM2 and EDM2 also showed similar premature polyadenylation and PolII elongation defects in TE regions (Fig. 4b, c; Supplementary Fig. 12c, 16), suggesting that rather than enhancing splicing of intragenic TE sequences, these factors likely suppress the usage of proximal APAs and Pol II release during the transcription of heterochromatic TEs in intragenic regions. These results demonstrate that epigenetic mechanisms impact TE-ATTS formation by regulating Pol II elongation and usage of APA in intronic or 3′ TE sequences.

## The presence of TEs in 3′-UTRs contributes to environmental responses and RNA stability

To understand the regulatory role of 3′-UTR TEs, we further investigated representative gene loci containing LTR/Copia superfamily retrotransposons in their 3′-UTRs; this TE superfamily showed overrepresentation in the TE-ALE events in the *Arabidopsis* genome (Fig. 2d). *GER5* (*AT5G13200*) responds to abiotic stress and phytohormones including abscisic acid (ABA), and is involved in reproductive development and regulation of seed dormancy[64]. The *GER5* locus has an insertion of *ATCOPIA78/ONSEN* in the 3′-UTR in an antisense orientation (Figs. 4a, 5a; Supplementary Fig. 13 and 19a)[44,48,49]. In addition to ATE-G isoform with TE-ATTS events, ONT-DRS also detected antisense transcripts of *GER5* promoted by ONSEN LTRs in Col-0 (Supplementary Fig. 19a). Quantitative PCR (qPCR) showed that *ibm2*, *edm2*, and *suvh456* caused transcription defects in the *ATCOPIA78* region and reduced transcription of the long ATE-G isoform spanning the *ATCOPIA78* (*MSTRG.32521.2*; Fig. 5b, c; Supplementary Fig. 19b). Instead, the mutants showed increased expression of short ATE-G isoforms terminated at proximal APA sites in the TE sequences (such as *MSTRG. 32521.4* and *.6*; Fig. 5c), showing an isoform switching of the ATE-G isoforms. ABA treatment induced higher expression of *GER5* as well as TE-G transcripts in wild-type Col-0 (Fig. 5b, c), while

*ibm2*, *edm2*, and *suvh456* showed higher expression of short ATE-G isoforms (*MSTRG.32521 .4 and .6*) compared with Col-0 (Fig. 5c). Interestingly, *Arabidopsis* accessions with no *ATCOPIA78* insertion in the *GER5* locus (Ler-0, Kyoto, An-1, and Sha) showed enhanced induction of *GER5* by ABA treatment compared with accessions with TE insertions (Col-0, Lan-0, Pna-17; Fig. 5d), demonstrating that the presence of *ATCOPIA78* represses induction of *GER5* transcripts, especially upon ABA treatment. Variable expression of the ATE-G isoforms was observed in the TE-containing Col-0, Lan-0, and Pna-17 (Supplementary Fig. 19c), suggesting additional ecotype-specific regulations of the transcripts. Since the presence of the TE in the 3′-UTR may also affect the stability of *GER5* mRNA[38], we examined its stability in Col-0 (TE + ), Ler-0 (TE-), Sha (TE-), *ibm2*, *edm2*, and *suvh456* by treatment with the RNA synthesis inhibitor cordycepin (Fig. 5e; Supplementary Fig. 20a). We found that *GER5* mRNA stability in wild-type Col-0 was comparable to TE-less accessions Ler-0 and Sha (Fig. 5e). However, *ibm2*, *edm2*, and *suvh456* showed a rapid reduction of *GER5* transcript levels after cordycepin treatment, suggesting that defects in the transcription of the ATE-G isoform spanning the *ATCOPIA78* resulted in *GER5* mRNA instability. Indeed, AU-rich elements[65] were prevalent in *GER5* transcripts terminated at proximal APAs in *ATCOPIA78* (Supplementary Fig. 20b), which may have led to enhanced degradation of *GER5* transcripts[38]. Thus, the presence of the TE in 3′-UTR of *GER5* contributes to responsiveness to ABA signaling as well as regulation of *GER5* mRNA stability.

Similarly, we examined the impacts of TE insertion in the *RPP4-ATCOPIA4* locus (Figs. 4a, 6a). *RPP4* is an *R* gene that encodes nucleotide-binding and leucine-rich repeat domains (NLRs), which mediates the effector-triggered immunity (ETI) response to isolates of the downy mildew oomycete *Hyaloperonospora arabidopsidis* (*Hpa*) Emwa1 and Emoy2[47,66]. In the *RPP4* locus of Col-0, *ATREP15* (Helitron) is inserted in the first intron, and *ATCOPIA4* (LTR/Copia) insertion in the sixth exon of the ancestral locus caused truncation of the 3′ part of the protein (Supplementary Fig. 13)[47]. In the combined Col-0 DRS data of this and a previous study[43], *RPP4* transcript isoforms coding for at least seven open reading frames (ORFs) have been detected (Supplementary Fig. 21–23). Interestingly, despite the ~ 4.7 kb of the *ATCOPIA4* TE insertion, the *RPP4* locus still generates transcript isoforms that preserve the ORFs of the original exons in the 3′ regions (annotated as *AT4G16857, CDS E* in Supplementary Fig. 21), encoding a putative C-terminal jelly roll/Ig-like domain (C-JID; Supplementary Fig. 21, 22). The C-JID domain might have a role in the pathogen recognition of RPP4[67], although RPP4 isoform without the C-JID region is sufficient for triggering ETI response[66]. Consistent with the previous reports[32,68], epigenetic mutants *ibm2*, *edm2*, and *suvh456* caused a loss of ATE-G isoforms spanning *ATCOPIA4* (such as *MSTRG.27750.7*; Figs. 4a, 6a, b; Supplementary Fig. 23a). Moreover, we found that ATE-G isoform associated with the Helitron *ATREP15* (*MSTRG.27750.20*) was also downregulated in epigenetic mutants (Fig. 6a, b; R-P5). Similar to *GER5* transcripts, *RPP4* transcripts in *ibm2*, *edm2*, and *suvh456* showed a rapid reduction after cordycepin treatment (Fig. 6c), suggesting that the production of ATE-G isoforms spanning the *ATCOPIA4* may affect the stability of *RPP4* mRNA (Fig. 6c; Supplementary Fig. 24). We further investigated the impacts of the ATE-G isoform production on the *RPP4*

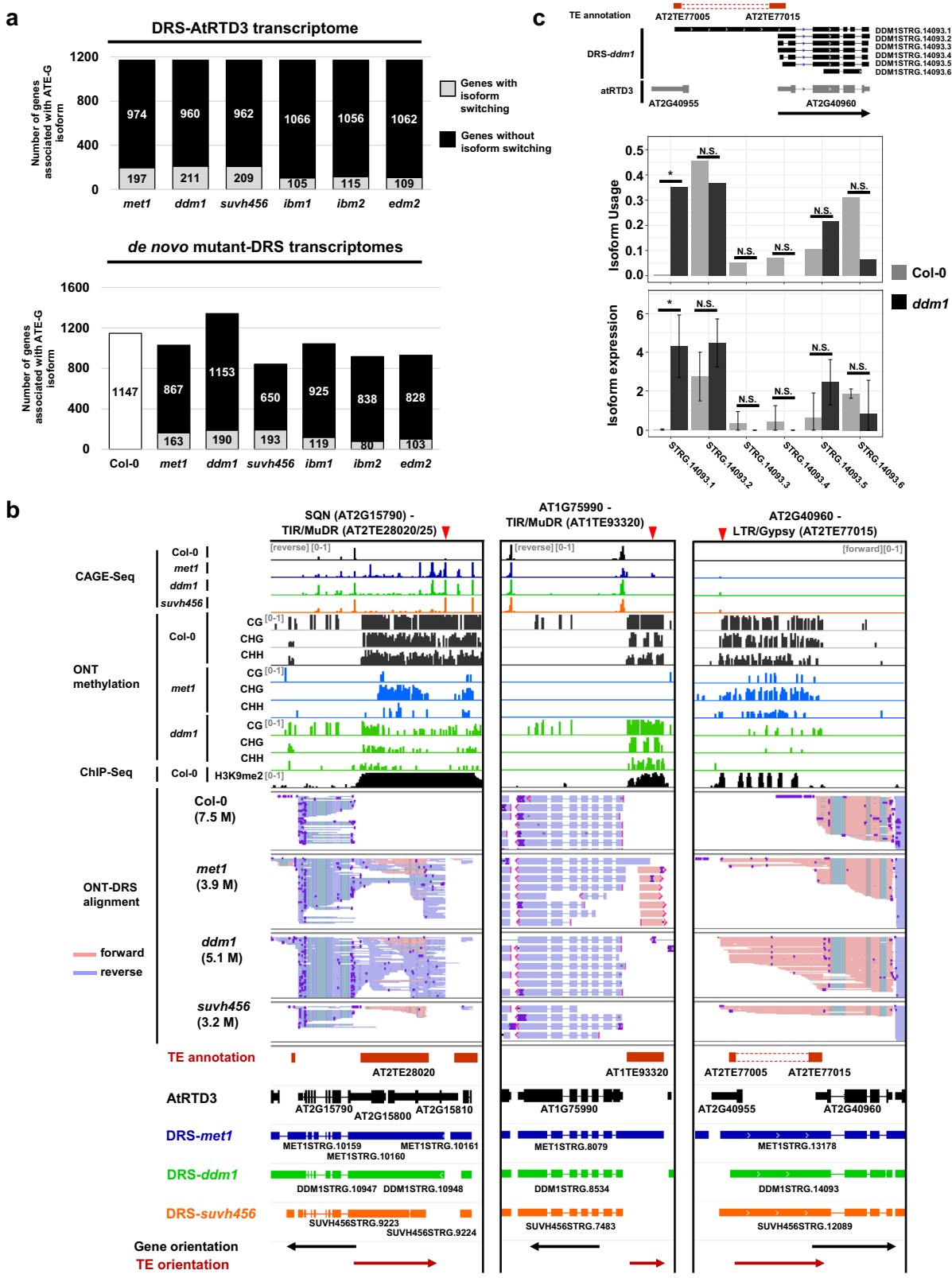

**a** DRS-AtRTD3 transcriptome

*de novo* mutant-DRS transcriptomes

function by assessing the ETI response to *Hpa*[47,69]. *Arabidopsis* natural accessions and epigenetic mutants were inoculated with *Hpa* isolate Emoy2, which is recognized by *RPP4* and elicits downstream responses[47]. *Arabidopsis* accessions NFA-10 and Kas-2, which are known to be susceptible to the isolate[70], showed strong infection symptoms as expected (Fig. 6d). Interestingly, *ibm2*, *edm2*, and *suvh456* showed increased ETI effectiveness against *Hpa* Emoy2

(Fig. 6d), which is in contrast to previous reports showing either a mild reduction in resistance or no differences in the mutants, respectively[32,68]. These differences in response may have originated from the tissues or isolates used in the study (true leaves in this study vs. cotyledon, for which has been reported age-related differences in resistance[71]; Emoy2 vs. Emwa1). To directly dissect the effects of TE insertion on the *RPP4*-mediated *Hpa* response, we also examined

**Fig. 3 | Epigenetic regulation of ATE-G isoforms. a** Top: Number of genes in each mutant showing isoform switching of ATE-G isoforms detected in DRS-AtRTD3. Bottom: Number of genes in each mutant showing change in isoform usage of ATE-G isoforms detected in mutant-DRS dataset. **b** Representative genome loci showing epigenetically regulated ATE-G isoform (Epi-ATE-G isoform) production with TE-ATSS events. Tracks (from top to bottom): CAGE-seq (reads per million. Only forward or reverse strands are shown); Col-0 ChIP-seq of H3K9me2 (bin per million); methylation level of each mutant in CG, CHG, and CHH contexts (0–100%); DRS read alignments of Col-0 and indicated mutants; TE and AtRTD3 transcript annotations, de novo assembly of transcripts in mutants, and the orientation of genes and TEs. Red arrows on the top indicate cryptic TSSs detected in epigenetic mutants. **c** Epi-ATE-G isoform production and isoform switching detected at the *AT2G40960* locus in *ddm1*. Top: *ddm1*-DRS transcripts aligned to the *AT2G40960* locus. Middle: Isoform usage (IF) of ATE-G isoforms. Benjamini−Hochberg false discovery (FDR) corrected *p*-values (*q*-values; *, *q* < 0.05) for isoform switching. Bottom: Expression levels of isoforms. Error-bars indicate 95% confidence intervals. Adjusted *p*-value from DESeq2 (*, *p*adj < 0.05).

T-DNA insertion lines in the *ATCOPIA4*. The SALK_005767 line showed susceptibility to *Hpa* (Fig. 6a, d), consistent with a previous report[72]. Interestingly, however, the second T-DNA line (SALKseq_035375) inserted downstream of *ATCOPIA4* showed a resistant response (Fig. 6d). qPCR showed that the SALK T-DNA insertions caused differential ATE-G isoform usage for *RPP4* under normal condition, such as different ATE-G isoform expression (R-P1 and R-P4; Fig. 6b), which may have contributed to the varying susceptibilities to *Hpa*. Although it is not clear how the T-DNA insertions in the TE sequence affect the overall *RPP4*-mediated pathogen response, the results suggest that the presence of the TE in the 3′ region of *RPP4* modulates the response to *Hpa* and the stability of *RPP4* transcripts.

## Environmental stresses affect alternative transcription termination by TE sequences

In addition to the epigenome modulation by mutants, we examined whether environmental stresses influence isoform switching and differential expression of ATE-G isoforms. Using public transcriptome data from biotic/abiotic stress treatment studies (ABA, methyl jasmonate, flagellin 22, salicylic acid, cold, heat, warm, salt, drought, and ultraviolet), we investigated changes in isoform usage and differential expression of representative ATE-G isoforms as well as mutant-specific Epi-ATE-G isoforms with TE-ATSS or TE-ATTS events (Figs. 3, 4; Supplementary Fig. 13–18). We found that Epi-ATE-G isoforms with TE-ATSS generated in a mutant background (*met1*, *ddm1*, and *suwh456*) were overall stably silenced under the stresses (Supplementary Fig. 25). In contrast, ATE-G isoforms and Epi-ATE-G isoforms with TE-ATTS detected in the epigenetic mutants (*met1*, *ddm1*, *suwh456*, *ibm1*, *ibm2*, and *edm2*; we also added the new RNA-seq dataset of *ibm12*, *ibm2-i7*[29], and *ibm2edm2*) showed changes in isoform usage and differential expression under various stresses, which often mirrored the responses of the epigenetic mutants (Fig. 7). Isoform switching and changes in expression manifested upon heat stress in gene loci with the heat-responsive *ATCOPIA78/ONSEN* insertion, associated with changes in DNA methylation in the loci[73] (Fig. 7; Supplementary Fig. 26). These data suggest that environmental signals can regulate the transcription and processing of ATE-G isoforms potentially via epigenetic regulation, which may contribute to adaptive responses to environmental changes.

## Discussion

In this study, we employed ONT-DRS technology to dissect the complexity of TE-G transcripts and their isoform production[39,40,42,43,49]. In addition, we developed the new bioinformatics tool ParasiTE to comprehensively detect ATE-G isoforms present in the *Arabidopsis* transcriptome. To our knowledge, ParasiTE is the first tool capable of identifying chimeric ATE-G isoforms and annotating associated RNA processing events (Fig. 1; Supplementary Note 1). The ParasiTE pipeline can be applied to a wide range of species using transcript, gene, and TE annotations as inputs and can be combined with short/long-read RNA-seq datasets to systemically identify ATE-G isoforms under various experimental conditions.

We found that about 3000 *Arabidopsis* genes are associated with TE-G transcript production, corresponding to about 8% of protein-coding genes annotated in AtRTD3 (Fig. 1d), which is close to an estimation made by a previous study[11]. In general, TEs tend to be short and degenerated in genic regions because of their deleterious effects on gene function[1,26]. Indeed, we found that TEs involved in ATE-G isoforms are shorter and less methylated than intergenic and intronic TEs (Fig. 2). Nevertheless, we still detected an enrichment of DNA transposon sequences in TE-AS events, associated with canonical splice donor-acceptor sequences (Fig. 2d, e). In animals and plants, TEs can provide cis-regulatory sequences for the expression of associated genes[2,6,23,35,74–76]. Thus, partial TE sequences could be co-opted as regulatory elements for transcriptional and post-transcriptional regulation of mRNA. On the other hand, young intact TE insertions are often polymorphic among natural accessions (Fig. 5)[77–80], suggesting that intragenic TEs may contribute to gene control for adaptation to the local environment and phenotypic diversity[81].

Our previous study revealed that thousands of cryptic TSSs in TEs are suppressed by repressive epigenetic modifications[23]. Similarly, we found that usage of cryptic APA sites within TEs in introns or 3′-UTRs was suppressed by repressive epigenetic marks (Figs. 3–6). We detected Pol II elongation and transcription of long read-through RNAs over several kb of heterochromatic TEs (Fig.4; Supplementary Fig. 12–18), indicating that heterochromatin per se is more permissive for Pol II transcription than previously considered. This process can be mechanistically separated from the IBM2/EDM2/AIPP1 pathway that suppresses the usage of cryptic APA in heterochromatic TEs since in *ibm2*/edm2, Pol II enrichment at the gene body of the *RPP4* locus associated with heterochromatic marks was not greatly affected (Fig. 4a), while Pol II signals sharply decreased downstream of cryptic APA in the 3′-UTR of *AT4G16860* in the mutants (Fig. 4a). In *Drosophila*, the Rhino-Deadlock-Cutoff (RDC) complex licenses Pol II transcription of piRNA clusters marked by H3K9me3[82,83]. The Cutoff protein in the complex suppresses the usage of APA sites in TEs, suggesting the conservation of a mechanism involving Pol II transcription of heterochromatic repeats[82,83].

In this study, we also investigated how TE-G transcripts are regulated by environmental signals. We focused on the *GER5* locus harboring the insertion of *LTR*/ATCOPIA78(*ONSEN*) in the 3′-UTR (Fig. 5). *ONSEN* has heat-responsive elements in the LTR regions and is activated in response to heat stress via the binding of heat-shock factors[84]. *ONSEN* insertions found in *GER5* and also *BEST1* (Supplementary Fig. 13) are considered as mobile and are highly expressed copies under heat stress[78]. Indeed, heat-shock experiments induced isoform switching of ATE-G isoforms in *GER5*, accompanied with a decrease in DNA methylation in *ONSEN* (Supplementary Fig. 26), showing that the *ONSEN* insertion confers heat responsiveness to the locus. *ONSEN* is known to introduce various transcriptional modulations to associated genes, including activation, alternative splicing, acquisition of heat responsiveness, and production of antisense RNAs[78,85]. Interestingly, we also found that the insertion of *ONSEN* causes a variable response of *GER5* transcription to ABA treatment in the natural accessions (Fig.5; Supplementary Fig. 19), which may be independent from the heat response and instead result from the instability of *GER5* mRNA caused by the TE sequence in the 3′-UTR (Figs. 4, 5). The instability seems to be mitigated in Col-0 under the normal condition by the production of the long ATE-G isoforms (Fig. 5d, e), while it became severe in s*uwh456* and *ibm2*/*edm2* (Fig. 5e). This suggests a potential role of the epigenetic machinery to minimize

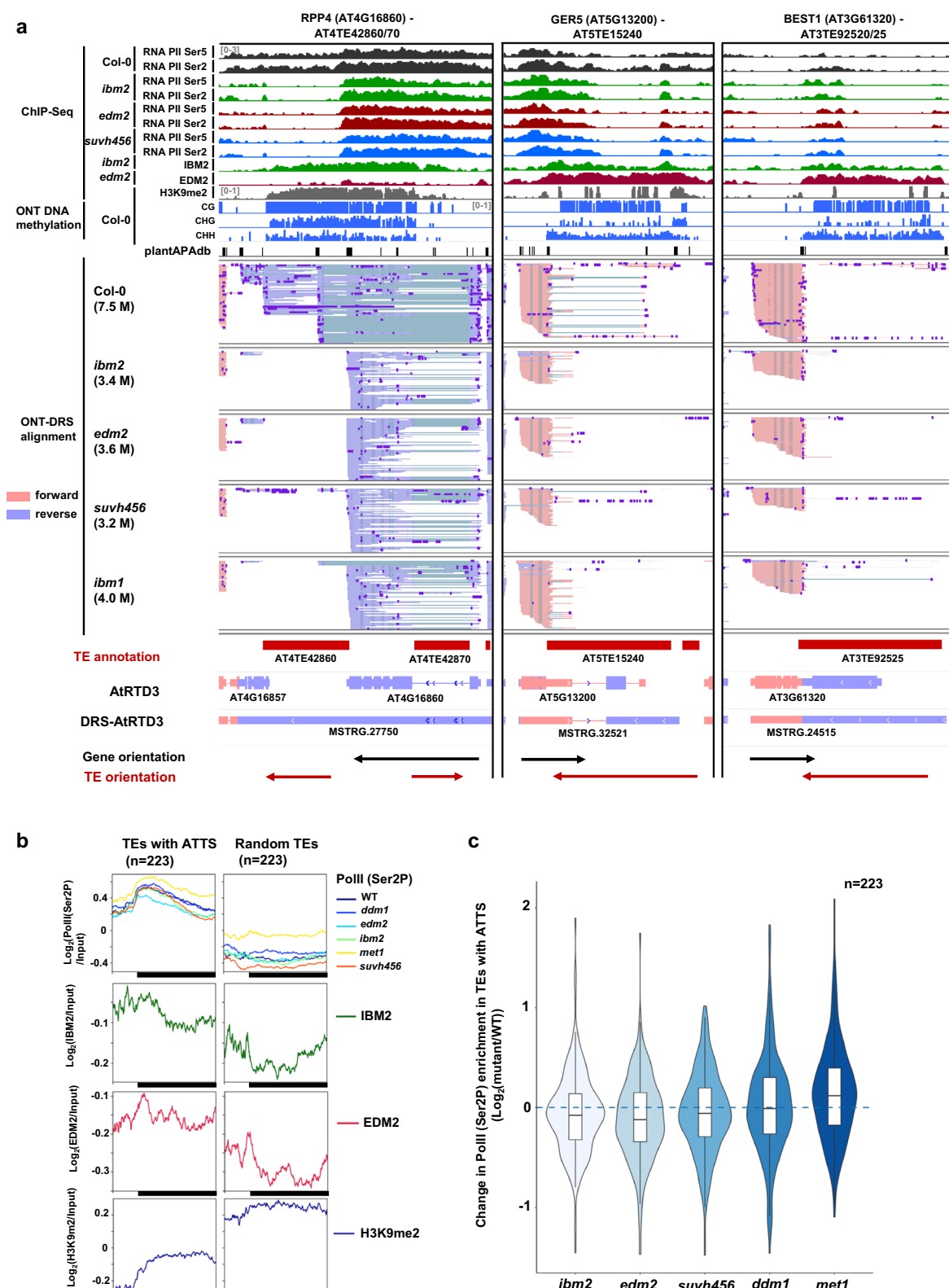

the impact of intragenic TE insertions on the host gene expression. TE sequences can provide AU-rich elements that may induce rapid degradation and turnover of the transcripts (Supplementary Fig. 20, 24), as has been observed in *FLOWERING LOCUS C* mRNA in *Capsella*, which results in variations in flowering timing[38]. Whether the insertion of *ONSEN* in the *GER5* locus and production of ATE-G isoforms are beneficial for environmental response and local adaptation

mechanisms remains unclear. We identified enhanced expression of the shorter form of *GER5* mRNA in epigenetic mutants, including *suvh456* (Fig. 5). Interestingly, a recent report showed that *GER5* is overexpressed with CHH hypomethylation in response to aphid feeding and is also constitutively activated in *suvh4/kyp* background, suggesting that *GER5* ATE-G isoform may contribute to the defense response and resistance to aphids[86].

**Fig. 4 | Epigenetic regulation of TE-ATTS. a** Representative genome loci showing Epi-ATE-G isoform production with TE-ATTS events. Tracks (from top to bottom): ChIP-seq data for RNA Pol II phosphorylated at Ser5/Ser2 in CTD repeats (bins per million); ChIP-seq data for IBM2 and EDM2 localization (bins per million); Col-0 ChIP-seq of H3K9me2 (reads per million); methylation levels of Col-0 in CG, CHG, and CHH contexts (0–100%); poly(A) sites obtained from the PlantAPA database; DRS read alignments of Col-0 and indicated mutants; TE and transcript annotations of AtRTD3 and DRS-AtRTD3 in this study and the orientation of genes and TEs.

**b** Metaplots for ChIP-seq signals of Pol II (Ser2P), IBM2, EDM2, and H3K9me2 over TEs with ATTS (isoform switching with $q < 0.05$ detected at least once among mutants; Supplementary Data 5; $n = 223$) or randomly selected TEs ($n = 223$). **c** Changes in Pol II (Ser2P) ChIP-seq signals in TEs with ATTS ($n = 223$) between the wild-type and mutants. The centerline represents the median. The borders of the boxplots are the first and third quartiles (Q1 and Q3). Whiskers represent data range, bounded to 1.5 * (Q3-Q1). Source data are provided as a Source Data file.

We found functional enrichment of pathogen responses including Cell killing and Defense response to fungus, in genes associated with TE-G transcript production (Fig. 1d). In plants, the ETI response to microbial pathogens is mediated by *NLR* genes, which often form gene clusters containing repetitive sequences and TEs[87]. *NLR* gene clusters are the most genetically and epigenetically divergent loci in *Arabidopsis* populations[88], and TEs contributed to the rapid evolution of *NLR* genes by enhancing recombination and exon shuffling[5]. We consistently found various ATE-G isoform transcriptions in the *R* genes *RPP4*, *RPP5-like*, *ADR1-L1*, and *RPP7* (Figs. 4, 6, 7; Supplementary Fig. 12, 15, 16). In these loci, TE sequences were often spliced out from mature mRNAs or introduced premature poly(A) sites, suggesting a minor contribution of TE sequences to the acquisition of novel protein-coding functions. In *Arabidopsis*, mRNA isoforms with premature termination codon are targeted for degradation by the nonsense-mediated mRNA decay (NMD) mechanism, which is often activated by stress-induced AS events[15,16]. Previous studies revealed that the NMD pathway targets NLR gene transcripts and is essential for tight control of immune receptor thresholds and suppression of auto-immune responses[89–91]. Thus, in addition to the epigenetic control of TEs surrounding NLR genes[63,89,92], transcripts with TE sequences might provide a signal for post-transcriptional regulation of NLR gene transcripts. In this study, we further identified the production of TE-G transcripts at gene loci with various functions in response to environmental signals (Fig. 7), suggesting mechanistic roles of TE-G transcript production in the regulation of mRNA processing and environmental responses.

## Methods
### Plant materials
*A. thaliana* mutants *met1-3*, *ddm1-1*, *ibm1-4*, *ibm2-2*, and *edm2-9* have been described previously[19,29,93–95]. *suvh456* seeds were kindly provided by Dr. Kakutani. All mutants were in the Col-0 background. The mutants *ibm1ibm2* and *ibm2edm2* were obtained by crossing *ibm1-4* with *ibm2-2* and *ibm2-2* with *edm2-9*, respectively. *ibm2-i7* is a transgenic line from the transformation of *ibm2-2* with the genomic DNA of *IBM1* without part of the sequence of the intron 7, as previously described[29]. Mutants were maintained as heterozygous for propagation, and second-generation homozygous *met1*, *ddm1*, *ibm1*, *ibm2*, *edm2*, *ibm1ibm2*, and *ibm2edm2* and T4 generation *ibm2-i7* were used for the RNA experiments described below. *suvh456* was maintained as triple mutants at least for three generations. The *ibm2-2* transgenic line complemented with the genomic DNA of *IBM2* fused with the FLAG-HA tag was previously described[29]. A transgenic line expressing EDM2-Myc-HA was obtained by transforming *edm2-9* with the genomic DNA of *EDM2* fused with MYC-HA tag sequence. The seeds were germinated and grown on 1/2 Murashige and Skoog (MS) plates under long-day conditions (16 h light; 8 h dark) at 22 °C, and seedlings at the 10-day-old stage were used for experiments.

### RNA extraction, Nanopore DRS and Illumina sequencing
For ONT-DRS (Oxford Nanopore Technologies, Oxford, UK) and Illumina short-read sequencing (Illumina Inc., San Diego, CA), 10-day-old whole seedlings of wild-type Col-0 and mutant plants were pooled for RNA extraction. Total RNA was extracted using RNAiso (TAKARA, Japan), and genome DNA was digested with TURBO DNase (Thermo Fisher Scientific, USA). For ONT-DRS, the Oligotex-dt30 <Super>

mRNA purification kit (TAKARA) was used for poly(A) mRNA purification. Purified mRNA from wild-type Col-0, *met1*, *ddm1*, *suvh456*, *ibm1*, *ibm2*, and *edm2* were used to generate libraries using the Direct RNA Sequencing Kit (SQK-RNA002, Oxford Nanopore Technologies) according to the manufacturer's instructions. Sequencing was performed using MinION device (device: MIN-101B, flow cells: FLO-MIN-106 R9 version; Oxford Nanopore Technologies) at the Okinawa Institute of Science and Technology Graduate University (OIST) Sequencing Center (SQC) or at our laboratory. We performed at least two sequencing runs for Col-0 and each mutant as biological replicates (Supplementary Data 1). To obtain a higher read depth for Col-0 ONT-DRS, previously generated public data for Col-0 ONT-DRS obtained by a standard ONT-DRS method (five runs) and by a 5′-cap capturing method (two runs) were combined with our Col-0 ONT-DRS data for detection of TE-G transcripts. Bases calling was performed with Guppy (v4.4.2.; https://nanoporetech.com/). Previously published short-read Illumina sequencing data (paired-end, 100 bp) for Col-0 and *ibm2*[26], and new RNA-seq datasets for *ibm1*, *edm2*, *ibm2-i7* (described previously[29]), *ibm1ibm2*, and *ibm2edm2* were obtained as described previously[26]. In addition, Illumina paired-end read data of *met1*, *ddm1*, *suvh456*, and Col-0 were retrieved from previous studies[23,29]. Illumina reads were trimmed using fastp (v0.21.0; parameters: -l 36 -r)[96]. Data are publicly accessible and the information are provided in Data Availability section.

### Processing DRS sequencing data
For ONT-DRS data analysis of the Col-0 and epigenetics mutants, we first converted raw RNA sequence data into cDNA sequences with seqkit (v0.12.1; option: seq−rna2dna)[97]. ONT-DRS reads from replicates were concatenated and then error-corrected using LorDEC (v0.9; parameters: -k 21, -s 3)[98] with the trimmed Illumina paired-end data of the corresponding genotype. Minimap2[99] was used to align corrected ONT-DRS reads to the *Arabidopsis* genome (TAIR10) retrieved from The Arabidopsis Information Resource (TAIR) (https://www.arabidopsis.org/) with parameters: -ax splice -G 10k -uf -k14, as previously described[40]. The percentage of Araport11 genes covered by at least 10 reads of ONT-DRS or Illumina RNA-seq reads shown in Supplementary Figure 6a, was obtained using featureCount (v2.0.2; options: -L for ONT-DRS and -p for Illumina reads).

### ONT-DRS-based de novo transcriptome analysis
A de novo transcriptome of Col-0 was built using Stringtie2 (v2.1.4)[100], as previously demonstrated[43,100]. Briefly, after processing of Col-0 DRS reads obtained from two biological replicates, the reads were combined with DRS data of Col-0 from a previous study (Parker et al. 2020 https://doi.org/10.7554/eLife.49658). The final DRS-Col-0 transcriptome annotation was generated using Stringtie2. We compared Stringtie2 "-R" or "-L" options using either Araport11[50] or AtRTD3[44] annotations as references (option -G). The "-R" option allowed to output unassembled, cleaned, and non-redundant long read alignments, and to collapse similar long reads alignments at the same location. This process resulted in a higher number of isoforms and was chosen to create the Col-0 DRS transcriptome with either Araport11 or AtRTD3 as references (Supplementary Fig. 1b and 1c). The AtRTD3 contains annotations of long non-coding RNAs genes, ribosomal RNA genes, transfer-RNA genes, micro-RNA genes in addition to the

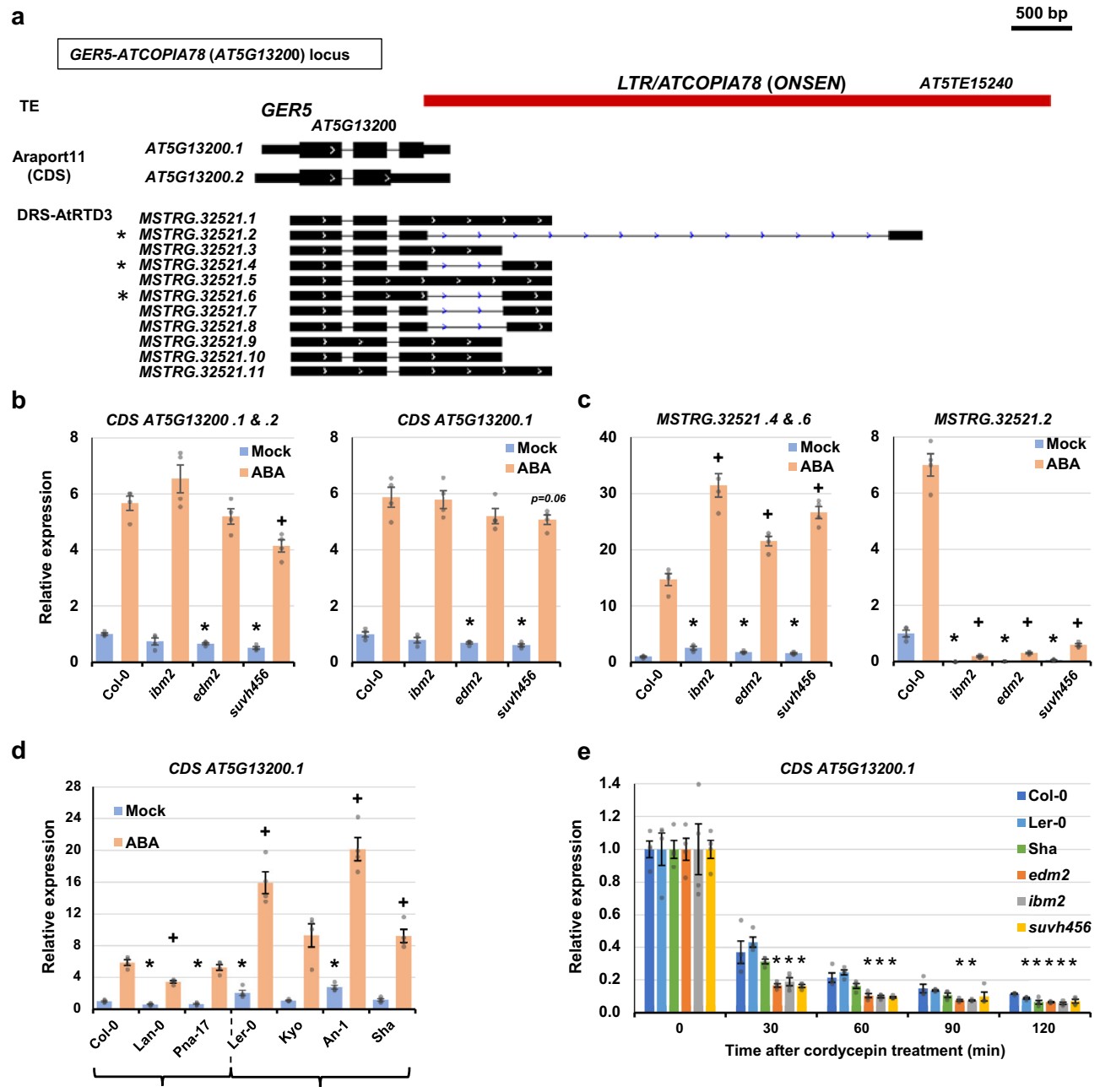

**Fig. 5 | Epigenetic regulation of ATE-G isoforms in the *GER5* locus and the impact on environmental responses and RNA stability. a** *GER5-ATCOPIA78/ONSEN* locus. TEs, Araport11 gene annotation, and DRS-AtRTD3 transcript isoforms are shown. * indicates isoforms examined by RT-qPCR in c. **b, c** Relative expression of transcripts corresponding to *GER5* protein coding sequence (CDS; *AT5G13200.1* and *AT5G13200.2*) and ATE-G isoform (*MSTRG.3521.2, 4, 6*) under mock and ABA stress conditions in indicated genotypes. Bars represent the means of four biological replicates ± standard error of the mean (SEM). *, *p* < 0.05 by *t*-test for comparison between Col-0 and mutants under mock conditions, and +, *p* < 0.05 by *t*-

test under ABA stress conditions. **d** Relative expression of transcripts corresponding to *GER5* CDS (*AT5G13200.1*) in the *A. thaliana* ecotypes with or without *ATCOPIA78/ONSEN* insertion in the 3′-UTR. Bars represent the means of four biological replicates ± SEM. *, *p* < 0.05 by *t*-test for comparison between Col-0 and mutants. **e** Relative transcript levels of *GER5* (*AT5G13200.2*) at 0, 30, 60, 90, and 120 min after cordycepin treatment in Col-0, *ibm2*, *edm2*, *suvh456*, and ecotypes without *ATCOPIA78/ONSEN* insertion (Ler-0 and Sha). Expression levels at 0 min are set as 1. Bars represent the means of four biological replicates ± SEM. *, *p* < 0.05 by *t*-test. Source data are provided as a Source Data file.

protein-coding genes. In order to improve the annotation of *A. thaliana* to facilitate the identification of ATE-G isoforms, we independently merged the two DRS transcriptomes (using Araport11 or AtRTD3 as a reference) with the corresponding annotations of Araport11 or AtRTD3 dataset using Stringtie2 (parameters:−merge -F 0 -T 0 -f 0 -g 1 -i -G) to generate DRS-Araport11 and DRS-AtRTD3 transcriptome annotations (Supplementary Fig. 1b). DRS-AtRTD3 was compared to the original AtRTD3 using GffCompare (v0.12.6)[101]. The

de novo transcriptomes of the epigenetics mutants were built using Stringtie2 (v2.1.4) with "-R" mode and the AtRTD3[44] annotation as the references (option -G). Data are publicly accessible and the information are provided in data availability section.

## ParasiTE: a new tool for prediction of TE-G transcripts
To detect TE-G transcript and ATE-G isoform productions, we developed a new tool named ParasiTE. This pipeline is composed

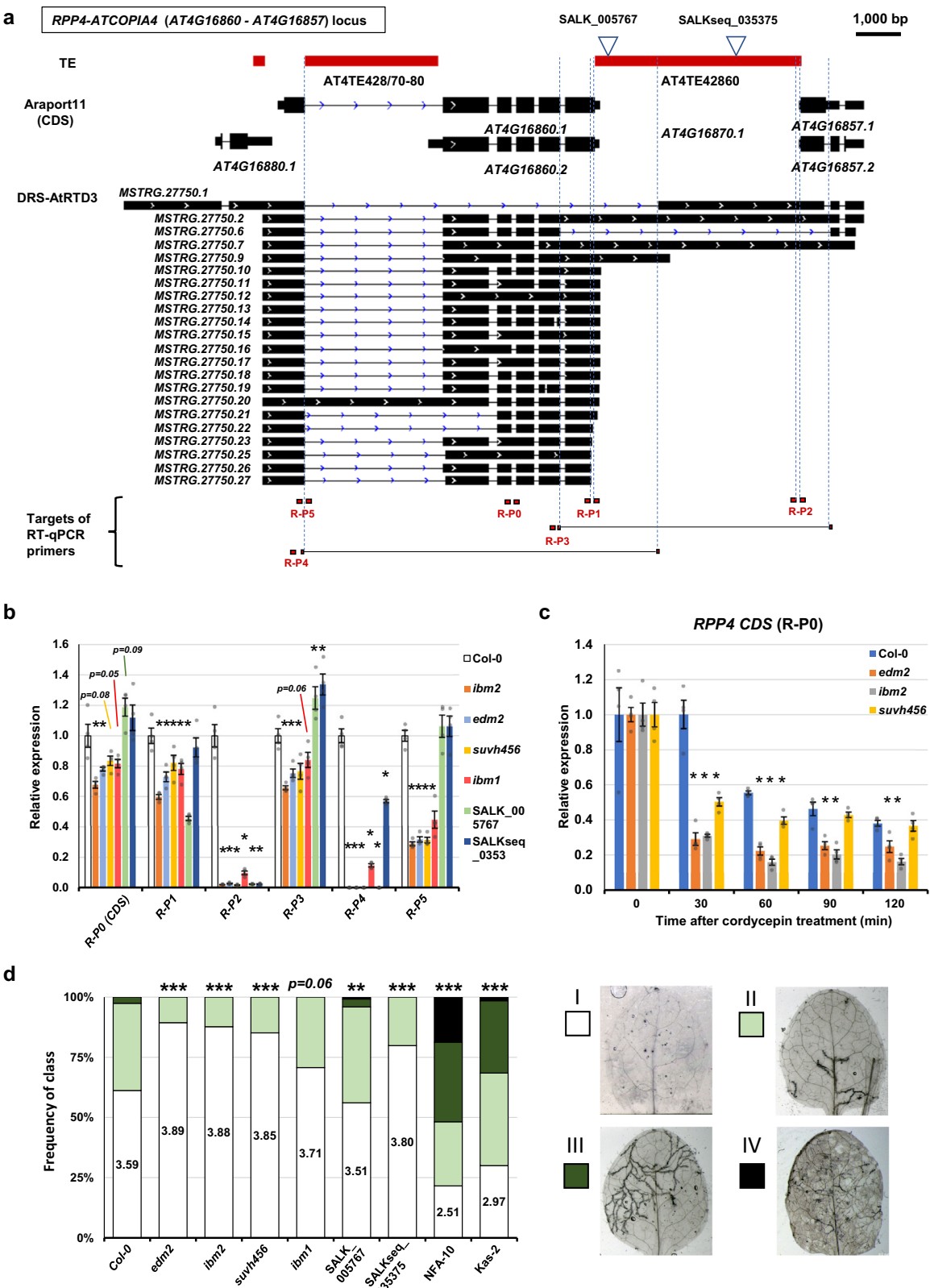

of five main steps: 1) removal of gene-like TE annotation and associated transcripts (annotation of TEs and associated transcripts included as genes in the gene annotation), 2) classification of intragenic and intergenic TEs, 3) classification of intronic and exonic TEs, 4) annotation of TE-G transcripts, and 5) annotation of TE-G transcripts with alternative transcript isoforms. Details of each step and additional filtering steps are described in Supplementary Note 1.

ParasiTE can be downloaded at (https://github.com/JBerthelier/ParasiTE).

### DNA extraction and Nanopore DNA sequencing

Ten-day-old whole seedlings of wild-type Col-0, *met1*, and *ddm1* were pooled for DNA extraction. Extraction of high-molecular-weight DNAs was performed using NucleoBond HMW DNA

**Fig. 6 | Epigenetic regulation of ATE-G isoforms in the *RPP4* locus and the impact on environmental responses and RNA stability. a** *RPP4-ATCOPIA4* locus. TEs, Araport11 gene annotation, DRS-AtRTD3 transcript isoforms, and primers for RT-qPCR are shown. **b** Relative expression of *RPP4* transcripts detected by RT-qPCR with primers indicated in a. Bars represent the means of four biological replicates ± SEM. *, *p* < 0.05 by *t*-test. **c** Relative transcript levels of *RPP4* at 0, 30, 60, 90, and 120 min after cordycepin treatment in Col-0, *ibm2*, *edm2*, and *suwh4S6*. Expression levels at 0 min are set as 1. Bars represent the means of four biological replicates ± SEM. *, *p* < 0.05 by *t*-test. **d** Incompatibility of *A. thaliana* ecotypes and mutants against *Hyaloperonospora arabidopsidis* infection. NFA-10 and Kas-2 are ecotypes

without the *RPP4* locus and were used as controls. Class I (white), hypersensitive response surrounding conidia penetration sites; class II (light green), presence of trailing necrosis in ≤50% leaf area; class III (dark green), presence of trailing necrosis in ≤75% leaf area; class IV (black), compromised ETI immunity, presence of pathogen hyphae not targeted by HR and conidiophores. Statistically significant differences in the frequency distribution of the classes between lines and Col-0 were determined by Pearson's chi-squared test; *, *p* < 0.05; **, *p* < 0.01; ***, *p* < 0.001. 70–130 leaves were analyzed per line across three separate experimental replicates. Source data are provided as a Source Data file.

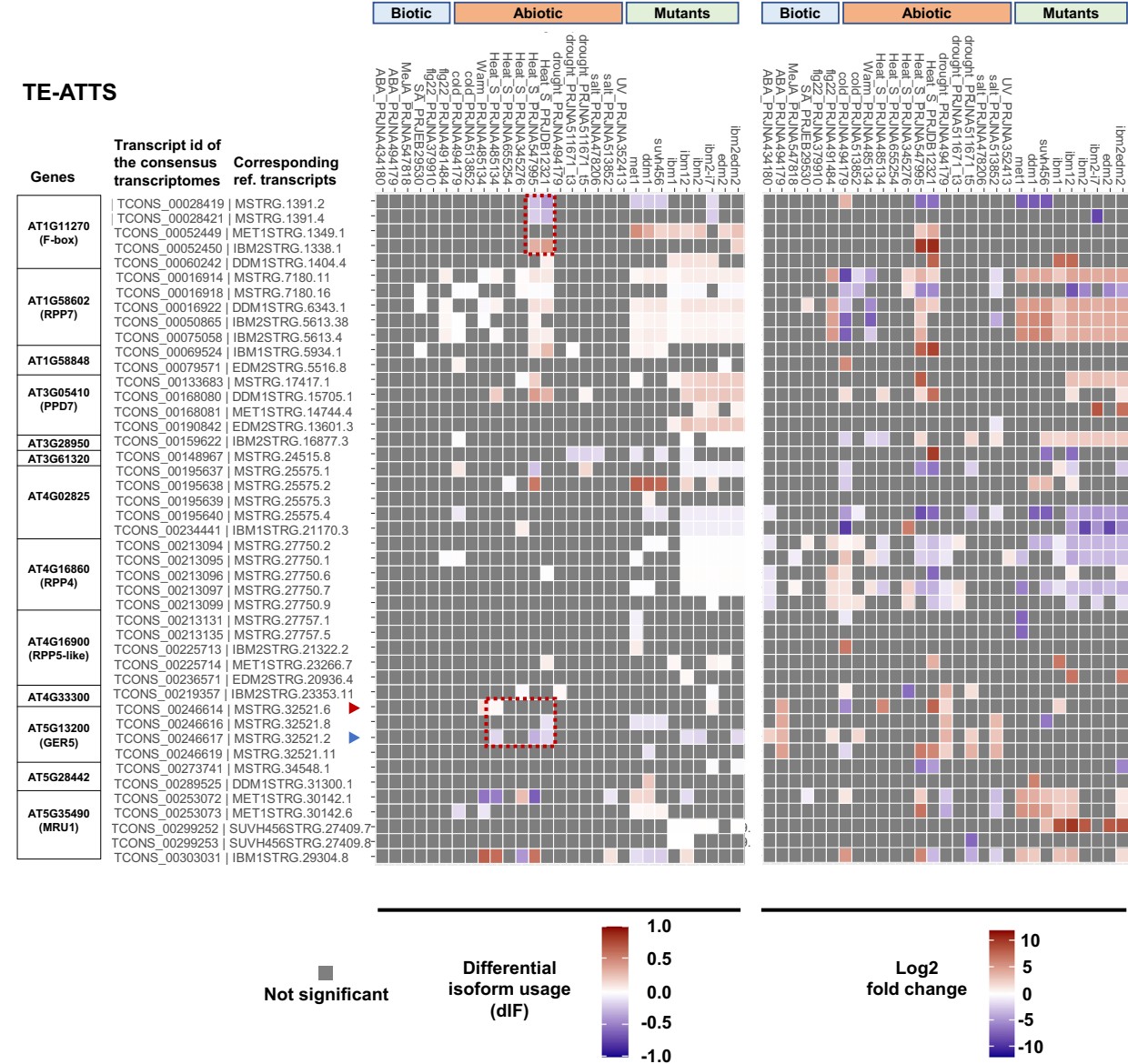

**Fig. 7 | Regulation of Epi-ATE-G isoform candidates under various stress conditions.** Heatmaps showing statistically significant differential isoform usage (dIF; left) and differential expression fold-change (right) with TE-ATTS under stress conditions or in epigenetic mutants. Transcriptome data with the stress treatment studies are from the public RNA-seq data. The Epi-ATE-G isoform candidate *AT1G58848* found by others was also added[32]. Red dotted lines highlight predicted

isoform switching at *GER5* (*AT5G13200*) and F-box gene (*AT1G11270*) loci in Col-0 under heat stress conditions. The names of the transcripts from the consensus DRS transcriptome as well as one corresponding to the reference transcript are indicated (more than one reference transcript from DRS-AtRTD3 or mutant-DRS are applicable to some transcripts). Source data are provided as a Source Data file.

(TAKARA). Library preparation was performed using the Ligation Sequencing Kit (Oxford Nanopore Technologies) and sequencing was performed using MinION (device: MIN-101B, flow cell: R9.4.1) at OIST SQC.

### DNA methylation calling from Nanopore and Illumina Bisulfite sequencing

Bases-calling of Col-0, *met1*, and *ddm1* was accomplished using Guppy (v4.4.2; https://nanoporetech.com/). DNA methylation at CG, CHG, or

CHH was called with DeepSignal-Plant[54]. Bisulfite sequencing of Col-0, *ibm1*, *ibm2* and *edm2* were performed with Post-bisulfite adaptor-tagging (PBAT) using Illumina sequencing as previously reported[23,26,102]. Bisulfite sequencing data of *suvh456*, as well as, of Col-0 in control and heat shock conditions were retrieved from previous studies[73,103]. Illumina reads were trimmed with fastp (v0.21.0; parameters: -l 36 -r)[96] and aligned to the *A. thaliana* genome (TAIR10) with Bismark (v0.23.0; parameters: -N 1 -X 1000−ambiguous -R 10−un−score_min L,0,−0.6)[104]. For PBAT Illumina reads, ambiguous and unmapped paired-end reads were mapped again as single-end reads and merged with the paired-end reads alignments. For each dataset, the unmethylated mitochondrial genomes were used as negative controls for the DNA methylation calling and to calculate the error rates. Estimated error rates were incorporated into a binomial test to assess methylation calling at each site, with q-value of <0.01[105]. Cytosine sites with non-significant calls were treated as unmethylated. Methylation levels were calculated using the ratio of mC/(mC + unmC), as previously indicated[106], and were converted to bedGraph format for visualization. We also calculated the percentage of DNA methylation of TEs with a minimum of 5 covered cytosines as previously described[27]. The difference in the percentage of TE DNA methylation in *met1* or *ddm1* compared with Col-0 was also calculated (Supplementary Data 3).

### GO term/TE enrichment and TE-RI splicing site motifs search

*Arabidopsis* gene IDs in the DRS-AtRTD3 transcriptome were retrieved from the original AtRTD3 data using gffcompare[101]. Gene IDs involved in TE-G transcript and ATE-G isoform events were submitted to ShinyGO (v0.75)[107]. For each TE superfamily, the expected number of TEs for each type of ATE-G isoform event was calculated with the following equation: Expected number of TEs belonging to a superfamily = ([Number of TEs in the superfamily]*[Number of TEs involved in the ATE-G isoform events])/Total number of TEs.

Fold-change = [Expected number of TEs belonging to a superfamily]/[Number of TEs belonging to a superfamily involved in the ATE-G isoform events]. *P*-values for under- or over-enrichment were calculated based on the cumulative distribution function of hypergeometric distribution. Annotations of donor and acceptor splicing sites, which overlap exons of transcripts involved in TE-IR events were retrieved using bedtools intersect function[108] (Supplementary Data 2). This step was performed independently for donor and acceptor splicing sites, but also for both transcript strands. Non-redundant donor and acceptor splicing sites were listed and a window of 4 nucleotides upstream and downstream was generated for each site. The splicing site windows overlapping with TEs involved in TE-IR events were kept as final candidates using bedtools intersect function[108]. Nucleotide sequences of donor and acceptor splicing sites were extracted using bedtools getfasta[108]. Weblogo[109] was used to visualize nucleotide enrichment of splicing sites associated with TE-IR events.

### ChIP-seq analysis

ChIP-seq data of RNA Pol II in wild-type Col-0 and the epigenetic mutants were obtained as previously described[23] and using anti-RNA polymerase II CTD repeat YSPTSPS (phospho S2, ab5095; Abcam, Cambridge, UK) and anti-RNA polymerase II CTD repeat YSPTSPS (phospho S5, ab5408; Abcam) antibodies. ChIP-seq for FLAG-HA-IBM2 and EDM2-MYC-HA was performed using an anti-HA antibody (ab9110; abcam). ChIP-seq peak of H3K9me2 in Col-0 and *ibm1* were retrieved from published data[57]. Raw sequence reads were trimmed with fastp (v0.21.0; parameters: -l 36 -r)[96] and aligned to the *A. thaliana* genome (TAIR10) with Bowtie2 (v2.4.2). ChIP peak calling was performed with MACS2 (v2.2.7.1; parameters:−broad−cutoff-analysis). Peaks overlapping with genes or TEs were extracted using the intersect function of bedtools (v2.29.2)[108]. For figures, normalized (log₂ [ChIP/input]) bigwig files were generated with the bamCoverage function of

deepTools (v3.4.3; parameters: −scaleFactorsMethod None -bs 10 −normalizeUsing BPM -e 150 −operation log2)[110] and visualized using Integrated Genome Viewer (IGV)[111]. Data are publicly accessible and the information are provided in Data Availability section.

### Isoform switching and differential expression analysis

Illumina short reads from RNA-seq were mapped on transcriptome assemblies with Hisat2[112] (v2.2.0; parameters: -p 100 −no-mixed −max-intronlen 10000). The tool Salmon[113] (v1.3.0) was used to perform the read count. Isoform switching under various stress conditions or in mutants was predicted with the pipeline IsoformSwitchAnalyzeR (v1.6.0)[114] employing DEXSeq[115] with difference in isoform usage (dIF) > 0.01 and isoform switch q-value < 0.05. The functions switchPlotIsoUsage and switchPlotIsoExp were used to investigate the *AT2G40960* locus (Fig. 3c). Differential expression of isoforms was detected with DESeq2[116] by importing the Salmon read counts with tximport[117].

### Identification of representative epigenetically regulated Epi-ATE-G isoforms

To identify Epi-ATE-G isoforms, we applied the following filtering steps to sort candidate ATE-G isoforms detected in DRS-AtRTD3 or in mutant transcriptomes by ParasiTE (Supplementary Data 5, 6). For Epi-ATE-G isoforms in *met1* or *ddm1*, we sorted Epi-ATE-G isoform candidates that were associated with reduced DNA methylation in the mutants. For Epi-ATE-G isoforms in *suvh456*, we searched for overlaps of ATE-G isoform loci (TE and transcript) with ChIP-seq signals of Col-0 H3K9me2. For Epi-ATE-G isoforms in *ibm1*, we searched for overlaps of ATE-G isoform loci (TE and transcript) with ChIP-seq signals of Col-0 H3K9me2 and *ibm1* H3K9me2[57]. For Epi-ATE-G isoforms in *ibm2* and *edm2*, we searched for overlaps of ATE-G isoform loci (TE and transcript) with ChIP-seq signals of IBM2, EDM2, as well as Col-0 H3K9me2[57]. Finally, candidate loci were manually filtered by visualization of mapped DRS reads, published CAGE-seq datasets of Col-0, *met1*, *ddm1*, and *suvh456*[23], and ChIP-seq signals and DNA methylation levels using IGV.

### Investigation of mutants and stress conditions on Epi-ATE-G isoform expression and isoform switching events

A consensus transcriptome was built by combining DRS-AtRTD3 and DRS-based mutant transcriptomes using GffCompare[101] (v0.12.6; -r with the AtRTD3 annotation as a reference). Next, Epi-ATE-G isoform candidates that were retrieved, as described above, were manually selected. A list and information about the RNA-seq data (short paired-end reads) used for the stress condition analyses are provided in Supplementary Table 5. For the heatmap presentation in Fig. 7, a maximum of five Epi-ATE-G isoforms are shown for each gene, and only ATE-G isoforms with at least one significant isoform switching event in an epigenetic mutant are presented. Because of the complexity of *RPP4* ATE-G isoforms, only the first five ATE-G isoforms in Fig. 6a are presented in the heatmap. Significant isoform switching (q < 0.05) was analyzed using IsoformSwitchAnalyzeR (v1.6.0)[114]. The function switchPlotIsoUsage was used to investigate the *GER5* and *F-box* loci (Supplementary Fig. 26). The transcripts in the heatmaps presenting isoform switching were also analyzed for differential expression with adjusted *p* value < 0.05 from DESeq2[116].

### Poly(A) site prediction analyses

Uncorrected ONT-DRS reads of Col-0 (unconverted to cDNA and uncorrected with LorDEC) were mapped to the *Arabidopsis* genome (TAIR10) with minimap2 (-ax splice -uf -k14). Nanopolish[118] was used to predict ONT-DRS reads with poly(A) tails. Bam files were converted into bed files, and for reads predicted to bear poly(A) tail, the poly(A) sites were assumed to be located at the 3' end of the read and at ±10 bp from the soft-clipped nucleotide (soft-clipping done by minimap2). The locations of TEs involved in TE-ATTS events predicted by ParasiTE

were compared to the APA predictions obtained with ONT-DRS and APA sites retrieved from the PlantAPAdb[62] (High confidence annotation; http://www.bmibig.cn/plantAPAdb/Bulkdownload.php, using bedtool's intersect function[108] (v2.29.2).

## Reverse transcription PCR (RT-PCR) and qPCR

For each experiment, at least three plants were pooled for a replicate, and four independent biological replicates were prepared for each experiment. Total RNA was extracted from 14-day-old seedlings (12 days old for the cordycepin experiments) using the Maxwell 16 LEV Plant RNA Kit and Maxwell 16 Instrument (Promega, Madison, WI). For RT-PCR and qRT-PCR, cDNA was synthesized using Prime Script II (TAKARA) following the supplier's protocol. RT-PCR was performed using GoTaq DNA Polymerase (Promega) using a T100 Thermal cycler (Bio-Rad, USA). RT-qPCR was performed using TB Green Premix ExTaq II (Tli RNAseH Plus) (TAKARA) with Thermal Cycler Dice ® Real Time System III (one cycle of 95 °C for 30 s followed by 40 cycles of 95 °C for 5 s and 60 °C for 30 s). Primer specificity was validated by a dissociation curve. *ACTIN2* (*AT3G18780*) and *GAPDH* (AT1G13440) were used as housekeeping genes, as previously recommended[119]. The $2^{-\Delta\Delta CT}$ method was used to determine mRNA expression levels[120]. All primers used in this study are listed in Supplementary Table 6 and 7.

## *RPP4* ORF prediction

*RPP4* transcripts (annotated from a to g in Supplementary Fig. 21, 23, 24) were predicted from the minimap2 alignment of corrected ONT-DRS reads of Col-0 to the *A. thaliana* genome (TAIR10). ORFs of transcripts were predicted using ORF Finder (https://www.ncbi.nlm.nih.gov/orffinder/), and CDS prediction was performed (annotated A to G in Supplementary Fig. 21). Multi-alignments were generated with ClustalΩ (https://www.ebi.ac.uk/Tools/msa/clustalo/) and visualized with Geneious (https://www.geneious.com). Domain and homologous superfamily predictions of corresponding amino acid sequences were obtained with InterProScan (https://www.ebi.ac.uk/interpro/search/sequence/).

## ABA treatment

Fourteen-day-old Col-0 and mutants were grown in a pot and 50 μM of ABA + 0.01% SILWET L-77 was sprayed onto the plants. Only water and 0.01% SILWET L-77 was sprayed onto the leaves of the mock plants. Biological replicates were harvested after 2 h of treatment and frozen in liquid nitrogen.

## Heat shock treatment

The plants were subjected to heat shock stress as described by Ito et al., 2011. Twelve-day-old Col-0 were grown in pots at 21 °C before being cooled at 6 °C for 24 h. Next, the control plants were moved back to the 21 °C cabinet while the others were subjected to heat shock treatment in a growth chamber (Biotron LPH-410SP; NK system, Japan) at 37 °C for 24 h. Biological replicates were harvested after treatment and frozen in liquid nitrogen.

## Cordycepin treatment of the *Arabidopsis* strains

Twelve-day-old Col-0, Ler-0, Sha, and epigenetic mutants were grown on 1/2 MS plates, and experiments were independently performed for each genotype. For each genotype, around 80 plants were transferred into a glass plate containing 80 mL buffer (1 mM PIPES, 1 mM trisodium citrate, 1 mM KCl, and 15 mM sucrose [final pH of 6.25]), as previously described[38]. The plants were incubated for 30 min in the buffer and covered with a transparent tissue layer that was engulfed by the solution. The plants were kept under light and were slowly shaken. Portions of the plants were collected after 30 min incubation; this corresponded to the time 0. Next, 20 mL of 3 mM of 3-deoxyadenosine (cordycepin; Thermo Fisher Scientific) was added to the buffer to

obtain a final concentration of 0.6 mM (150 mg/L), and vacuum infiltration was performed for 30 s at 0.04 MPa. The plants were collected at time points t = 30, 60, 90, and 120 min after the start of the treatment and frozen in liquid nitrogen. RT-qPCR was performed as described above. The expression levels of genes at each time point were first normalized using the expression levels of *ACTIN2* and *GAPDH* and then normalized to the expression levels at time 0. Eukaryotic initiation factor-4A (EIF4A1, *AT3G13920*) and EXPANSIN-LIKE A1 (EXLA1, *AT3G45970*) were used as control genes for high and low mRNA stability, respectively, as previously described[38,121]. Primers and the efficiency of primers used for this experiment are reported in Supplementary Table 6.

## *Hpa* resistance assays

Inoculation with *Hpa* Emoy2 was accomplished as described previously[122]. Briefly, the *Arabidopsis* plants were spray-inoculated to saturation with a spore suspension of $1 \times 10^5$ conidiospores/mL of Emoy2. Plants were covered with a transparent lid to maintain high humidity (90–100%) conditions in a growth cabinet at 16 °C under a 10-h photoperiod until the day of sampling. To evaluate hyphae growth and dead cells, leaves inoculated with Emoy2 were stained with trypan blue. Infected leaves were transferred to trypan blue solution (10 mL of lactic acid, 10 mL of glycerol, 10 g of phenol, 10 mL of $H_2O$, and 10 mg of trypan blue) diluted in ethanol (1:1 v/v) and boiled for 1 min. Leaves were destained overnight in chloral hydrate solution (60% w/v) and then stored in 60% (v/v) glycerol.

Trypan blue-stained leaves were analyzed with a stereomicroscope (KL200 LED; Leica, Schott, Germany), and assigned to different classes based on Emoy2 interaction: class I (white), hypersensitive response surrounding conidia penetration sites; class II (light green), presence of trailing necrosis in ≤50% leaf area; class III (dark green), presence of trailing necrosis in ≤75% leaf area; class IV (black), compromised ETI immunity, presence of pathogen hyphae not targeted by the hypersensitive response (HR) and conidiophores. Statistically significant differences in the frequency distribution of the *Hpa* colonization classes between lines and Col-0 were determined by Pearson's chi-squared test. Between 70 and 130 leaves were analyzed per line across three separate experimental replicates.

## Statistics and reproducibility

Plants were placed randomly in the plant facility. No statistical method was used to predetermine sample size. No data were excluded from the analyses. No blinding was applied for sampling.

## Reporting summary

Further information on research design is available in the Nature Portfolio Reporting Summary linked to this article.

## Data availability

The sequencing data generated in this study have been deposited in EMB-EBI European Nucleotide Archive database under accession codes PRJEB53848 (Illumina RNA-seq data, note that the corresponding Col-0 and *ibm2* dataset were previously published[26]); PRJEB53877 (ChIP-seq data of epigenetics mutants); PRJEB53877 (ChIP-seq data of RNA Pol II in *ibm2* and *edm2*); PRJEB53881 (ONT-DRS data of epigenetics mutants); PRJEB53882 (ONT DNA of Col-0, *met1*, and *ddm1*); PRJEB58752 (Bisulfite-Seq data of Col-0, *ibm1*, *ibm2* and *edm2*). The processed ONT-DRS transcriptomes, ONT methylation data, and ChIP-seq data are available at https://plantepigenetics.oist.jp/. Source data are provided with this paper.

## Code availability

ParasiTE is available in Github https://github.com/JBerthelier/ParasiTE or https://doi.org/10.5281/zenodo.7820890.

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

## Acknowledgements

This work was supported by MEXT Grant-in-Aid for Transformative Research Areas (A) JP20H05913 to H.S., JP20H05909 and JP22H00364 to K.S., JP20H02995 to S.A., and by OIST. We thank the Arabidopsis Biological Resource Center and the Salk Institute Genomic Analysis Laboratory for providing *Arabidopsis* T-DNA insertion mutants, OIST SQC for Nanopore DRS-seq and DNA-seq, Illumina RNA-seq, ChIP-seq, and BS-seq sequencing services, Dr. Tetsuji Kakutani for providing mutant seeds, Dr. Tu Le and OIST IT Section for technical supports in building the web interface to access data, as well as OIST English editing service for proofreading of the manuscript.

## Author contributions

DRS experiments were designed by J.B. and H.S and performed by T.S., J.B., and H.S. mRNA expression experiments were designed by J.B., H.S., and L.F. and performed by J.B. with the help of L.F. Pathogen experiments were performed by S.A., K.S., and L.F. ParasiTE was developed by J.B. Data analysis was performed by J.B. and ONT methylation calling were performed by J.B. and M.S. The manuscript was prepared by J.B. and H.S.

## Competing interests

The authors declare no competing interests.
