## [Peer Review File · Nature Communications]

Long-read direct RNA sequencing reveals epigenetic regulation of chimeric gene-transposon transcripts in *Arabidopsis thaliana*REVIEWER COMMENTS

Reviewer #1 (Remarks to the Author):

In this manuscript, the author provided a global characterization of TE-gene transcripts (TE-Gts) in Arabidopsis by using Nanopore direct RNA-seq and the self-developed bioinformatics pipeline "ParasiTE". They identified the enrichment of TE superfamilies associated with different TE-Gt alternative isoforms. By analyzing the epigenetics modification, they found that the heterochromatic marks were required for Pol II transcription and impacted TE-ATTS (alternatively transcription termination sites) formation. Lastly, they investigated the environmental response to TE-Gts. Overall, this study provides a rich resource of chimeric gene-transposon transcripts in Arabidopsis. And combining various epigenetic mutant data, they illustrated the association between epigenetic state and TE-gene interaction, providing a new angle for studying transcription regulation. The comments below are intended to strengthen the study.

Major comments:

1. The authors detected a total of 11,348 TE-Gt and 6,025 altTE-Gi events. Did they consider the read coverage bias? Since the pore block and the 5'-3' degraded RNA transcript can result in truncated reads, leading to a 3' bias of ONT-DRS results (PMID: 32033565). Such bias may lead to an overestimation of isoforms.
2. It is surprising that heterochromatic modification, such as H3K9 and DNA methylation, were required for Pol II elongation in some specific loci. From Fig.4a, the 3' terminal of long readthrough read seems coincident with PAS. Is it possible that this observation is associated with the choice of different PAS?
3. Fig3a and Fig3b showed that there are differences in the number of altTE-Gis among mutants based on the short-read RNA-seq datasets and DRS data. The author explained the smaller number of altTE-Gis due to the low coverage among other mutants. While the higher number of altTE-Gis occurs in ddm1, suggesting the dramatic impact of ddm1. So if the coverage of the data increases, is it possible to see changes in other mutants?
4. How the author explains or conjectures that in met1, the pol II enrichment increased(Fig. 4c). It is not mentioned in the paper.
5. The DRS raw data in fast5 format should be deposited as well.

Minor comments:

1. The DRS-AtRTD3 annotation should be provided either as a supplementary file or via the website <https://plantepigenetics.oist.jp/>.
2. Line 215, 'H3K9m' the description is not entirely correct. H3K9 methylation or the abbreviations 'H3K9M' is more appropriate. And in line 271, the word changes to 'H3K9'. Please contextualize.
3. Fig. 4b, the line color of different mutants was not annotated. It is difficult to correspond with each other.
4. There seems to be something wrong with the correspondence of the figure to the article, such as in line 408, please check again.

Reviewer #2 (Remarks to the Author):

Berthelie and colleagues use ONT DRS in Arabidopsis mutants linked to epigenetic effects on the expression of TEs. The authors develop a computational pipeline focused on TEs. Since TEs are often repetitive in sequence ONT DRS with longer reads offers an advantage over short-read NGS methods. Specifically, the authors generate ONT DRS data for heavily used mutants in the field (met1, ddm1, suv456, ibm1, ibm2, edm2). These data are of good

quality and the authors performed 2 repeats, but the read depth is rather low. The low read depth required the authors to rely on previous ONT DRS data to increase read counts for wild type that was already utilized for similar approaches. The authors identify a range of scenarios how RNA derived from TE sequences can interact with mRNA expression near the TE. The authors integrate analyses for some loci where the interactions of TEs with mRNAs have functional effects and were studied in detail previously. Overall, I believe that some of the data and analyses tools will be of interest to the TE community. The manuscript is very dense with acronyms, which makes it suitable for a specialized audience. The manuscript covers a lot of ground, which I appreciate since it is ambitious, yet here some conclusions need to be phrased more carefully as some conclusions have more the character of hypotheses. The authors also missed the windfall benefit of important studies in the field where some of the findings are already in the data. All in all, the manuscript offers some interesting aspects. The strength of the manuscript is that the associated data could be a useful resource, but here I am unsure if the read depth will be sufficient to be taken up as such by the community.

Comments:

- The authors need to make the manuscript more readable. The authors use far too many acronyms. Many of the acronyms are not defined, or it is not presented when first mentioned. What does ParasiTE stands for? TE-IR, TE-ES, TE-Gts, altTE-Gi, TE-AS, TE-ATTS etc need to be defined at first mention. A new, high quality figure that clearly explains all the acronyms is warranted, but they need to reduce the acronym density.
- By looking at the examples in this MS and data from other sources I got the impression that at least a fair fraction of the transcripts that the authors find in chromatin mutants are in fact already detectable in nuclear RNA degradation mutants. I find this is an important conceptual point the authors have missed, it challenges the hypothesis that the transcript are “only” detectable in the chromatin mutants. Yes, expression is probably induced as expected and often already shown, but many isoforms can be detected as “cryptic RNAs” without disrupted chromatin repression (PMID: 32444691, 31863587).
- There are contradictions in the presentation of novelty: “limitations of short-read RNA sequencing and a lack of bioinformatic pipelines have hindered the detection and comprehensive analyses of these transcripts in plants.” Lines 91-92. Then: “We found that about 3,000 Arabidopsis genes are associated with TE-Gt production, corresponding to about 8% of protein-coding genes annotated in AtRTD3 (Fig. 1), which is close to an estimation made by a previous study.” Lines 383-385. If they find the same number of these transcripts, there surely must have been already a method to identify them.
- How were the two ONT DRS repeats treated? Like true repeats, as in – the tracks only show RNA isoforms consistent in both repeats – or, where the repeats just merged? In case the repeats were merged without the removal of reads that support isoforms by some singleton jackpot reads specific to one of the repeats, the benefit of the two repeats is somewhat diminished. The authors should be clear about how they treated the repeats in the main text. I got the impression that the low depth of their ONT DRS data may have forced the authors to merge the reads of their repeats (and use prior data for wt).
- it would have been good to see if WT has been included in the Fig. 3a to compare the isoforms in WT to the mutants.
- How big is the overlap of their transcript isoform detection method and other approaches? For example data-driven transcriptome reconstruction (PMID: 34058980) and alternative RNA isoform detection methods (PMID: 32444691)?
- It would have been good to see if there was any comparison made between this tool and existing tools to show the advantages of ParasiTE over them.
- The ATSS part can be enhanced by providing context, key publications that address the roles of chromatin on the regulation of intragenic TSSs (PMID: 30707695, PMID: 32863101).
- CAGE data and ONT should be scaled, normalized (page 53, b)
- Fig2b-at2g15790-transcripts are already visible in plaNET-seq data (PMID: 31863587). Fig2c should be at2g40955.

- Fig4a, at4g16860 and at5g13200 (also Fig. 5a) already in TIF-seq data (PMID: 32444691), especially hen2 mutant TIF-seq data. Fig4a, at5g13200 even already called as MC transcript (PMID: 34058980).

- citations 68, uses the author 1st name instead of last name.

- How are the authors propagating the chromatin mutants? Are they regenerating heterozygotes so that we can be reasonable confident that we are looking at real effects of the epigenetic mutant rather than composite effects of increasingly deteriorating epigenetic indirect effects and the direct epigenetic effect in these mutants? A degron strain for rapid protein depletion of the factors would be a much better experimental approach.

Reviewer #3 (Remarks to the Author):

Key results: Bertheliet et al. developed a pipeline that leverages long-read transcriptomics to evaluate the contribution of transposable element (TE) sequences to the diversity of gene transcript isoforms. ONT-DRS data from multiple mutants show how gene-TE transcript genesis depend on maintenance of heterochromatic marks and on IBM2 and EDM2, factors required for heterochromatin transcription. Importantly, the authors highlight how selected alternative gene transcripts arise from recent TE insertions which ultimately influence mRNA isoform stability and gene responsiveness to environmental cues.

Significance: The study is highly original, provides important insights in deciphering the role of TE polymorphisms into environmental cellular response and potentially, local adaptation. The authors made the effort of designing a bioinformatic pipeline adaptable to other datasets and organisms, which is a significant contribution to the scientific community.

Data & methodology: Experiments showing the physiological relevance of the TE-gene transcripts were carefully controlled and conducted. Data analysis and representation has been well conducted, with some exceptions detailed below. The supplementary note 1 explains well the authors methods and strategic choices for the development of their design, showing a thoughtful design. Statistical tests are appropriately used.

Validity: The study suffers from minor flaws that should be addressed but do not affect the overall conclusions of the study. Most importantly, the manuscript's exciting biological insights are concealed by a lack of conciseness and an overuse of jargon, numbers and abbreviations in the result section that will discourage the reader, especially in the results section. This is particularly striking as the discussion is well written. The authors should try to clarify their message and improve the writing of the result section.

I recommend the editor to invite the authors to revise their manuscript before acceptance.

Major points:

1) Some of the reported phenomena are not quantified or described sufficiently. Figures 4, 5 and 6 and related supplementary figures are mostly based on genome browser snapshots that provide convincing examples, but the authors did not quantify some of these observations even though they have the data in hands.

Magnitude (line 213): "10–20 % of altTE-Gis showed significant differential isoform usage". This is a key result that the authors should describe in more details. What is the magnitude of the effect? Authors could show the distribution of isoform fold changes and inform the reader about the threshold used here for significance.

Correlation with changes in mDNA & H3K9me2: in "Alteration of epigenetic modifications impacts TE-Gt production", authors analyzed chromatin mark changes only for selected gene snapshots. They should plot changes in heterochromatic marks for all TEs associated with differential isoform usage in the mutants. This would help determine if isoform switching is a direct consequence of the loss of heterochromatic marks or not.

Quantification: in "Epigenetic regulation of TE-ATTS formation in *A. thaliana*", changes in TE-ATTS in the mutants have not been quantified by the authors. This is needed to give a reader a sense of how much TE-ATTS contributes to the 10-20% of altTE-Gis with significant

isoform switching in the mutants. Similarly, among TE-ATTS events, how often the mutants show premature termination, and how often do they show early termination? How does that correlate with changes in epigenetic marks? Most of this paragraph is based on genome browser snapshots, with convincing examples, but the reported phenomena must be quantified. Same observation for TE-ATSS. One could guess from fig7 and supp fig19 that these numbers are small, but if that is the case, authors should explicitly report them and comment.

2) Overinterpretation

- "DRS of the Arabidopsis transcriptome"

Line 172-173: "we found that TE sequences associated with TE-Gts or altTE-Gis were mainly located in 3'-UTRs of genes". This conclusion is not supported by the data. Supp fig2b shows that TE-Gt & altTE-Gi are more frequent in 3'-UTR than CDS & 5'-UTR, but the difference is small and the phrasing should reflect this. The current phrasing is misleading.

- "Contribution of TE superfamilies to TE-Gt production"

This part of the results is not very convincing. The authors report some minor over and under-representation of TE superfamilies and families. The fold change of observed / expected are not striking (fig 2a, 2d). I would rather conclude that all superfamilies contribute to TE-Gt, TE-AS and TE-ATP, with some superfamilies / families contributing more to the phenomenon. For example, the fold change enrichment for TE families in fig 2e) is of 1.63 for ATIS112A, 2.70 for ATENSPM1 + ATENSPM1A (supp data 3). Those are not very high numbers. The ATMU6N1 family shows the strongest enrichment (13.31, supp data 3), why not show the splicing donor / acceptor sites for this family?

More importantly, a canonical splicing donor site is GU, while an acceptor site is AG. Did the authors observe different consensus sequences with other TE families? Or when they merged all families together? If that is the case, then they could indeed conclude that "Overall, these results suggest that enrichment of particular TE superfamilies in intragenic regions contributes to splicing events of a subset of genes.". In the current state, there is no evidence to support this conclusion.

What can be concluded from fig 2b? This plot likely reflects the difference in size of transposon superfamilies themselves. One would have to look at the length of TEs associated to altTE-Gi after normalizing to the superfamily average size. The plot in its current state does not provide any valuable information. Therefore, the sentence "Most of the TE sequences associated with altTE-Gis were short and truncated, while retrotransposons (Copia [RLC], Gypsy [RLG], and LINE [RIX]) superfamily sequences were relatively long when compared with other TE superfamilies (Fig. 2b)." (lines 183-185) should be removed / reworked. The proper way to look at TE size would be to ignore TE superfamilies, or to plot the data as the authors did on fig S3c.

- "Environmental stresses affect alternative transcription termination by TE sequences"

"These data suggest that environmental signals can regulate the transcription and processing of altTE-Gi via epigenetic regulation, which may contribute to an adaptive response to environmental changes." The authors did not provide a single piece of evidence that the loci showing altTE-Gi were epigenetically modulated by environmental stresses. Stresses are known to affect splicing patterns, and there is no evidence to my knowledge that there is an epigenetic basis to this phenomenon. If authors really want to make this point, they will at least have to show that alternative isoform usage at TE-Gt upon stresses correlate with changes in chromatin marks. The data is still very interesting as such, showing that TE insertions can affect transcript isoform stoichiometry in response to environmental perturbations.

Minor points:

1) Rework the first four subsections of the results section (lines 106 to 204)

These four subsections are unappealing and should be reworked.

- Only methods that are relevant to data interpretation should be explained in the results, other details should be moved to the Methods section or if necessary, to the figure legends.

- The authors constantly refer to DRS-AtRTD3 and DRS-Araport11. The authors made it clear that DRS-AtRTD3 was their reference of choice in the 1st subsection, so they can spare this detail. Whether any number comes from an analysis with one reference or the other belongs to the figure labels or the figure legends.
- About "Identification and classification of TE-Gts in *A. thaliana*": this subsection has too many numbers to be readable. All numbers are already shown in figure 1, percentages are enough for the text and will improve readability.
- What information provides the following sentence (line 156): "27,628 genes were associated with Arabidopsis Genome Initiative [AGI] codes"? Remove or explain to the reader why this is relevant.

Abbreviations should be removed as much as possible and reworked. The diversity of splicing events requires the extensive use of abbreviations, therefore an effort should be made to avoid other abbreviations, especially when they are used only once. I suggest removing: AREs (used once in the text), CTD (once), BP (once in fig1), CC (once in fig1), MF (once in fig1), AGI (three times), dIF (once in fig7), possibly others that I missed. Abbreviations for TE superfamilies seem only relevant to give more space to figures 2b and 2d. I suggest getting of them entirely and reworking the figure. A horizontal barplot would give more space to the figure for example.

More importantly, abbreviations should be more intuitive to interpret:

- "Alternative" is sometimes abbreviated with "A" (APA) and sometimes with "alt" (altTE-Gis), this is confusing.
- The abbreviations combining consecutive uppercase and lowercase letters to distinguish consecutive words are very unintuitive, making the reading quite unpleasant. Here are my propositions for improvement:
 - o TE-Gts: TE-G transcripts or TE-G-Ts
 - o altTE-Gis: ATE-G isoforms or A-TE-G-Is
 - o Epi-altTE-Gi: Epi-A-TE-G isoforms

2) Does the figure of 39,998 gene models include lncRNA genes, ribosomal RNA genes, transfer-RNA genes, micro-RNA genes, etc.? Or is it only protein-coding genes? This information should be provided to the reader (I didn't find it in the methods).

3) Why does the data aggregation of DRS-AtRTD3 detect 20% more genes with TE-Gts than Araport11 and AtRTD3? Could it be that overall transcript coverage is increased because more tissue types are involved when they combine datasets? Or is it simply because the authors sequenced deeper?

4) Supp fig3b: how does BS-seq detect DNA methylation in such a repeated region? If this region is repeated and unmappable uniquely by short-read sequencing, the only way to measure DNA methylation there would be to include multi-mapping reads, is that the case here? If not, there should be no signal at all in BS-seq data. Below is the same region with uniquely-mapped reads from BSseq data:

5) In "Epigenetic regulation of TE-ATTS formation in *A. thaliana*": the presence of the TE insertion induces instability (the CDS transcript is less upregulated upon ABA treatment, fig 5d), and the distal APA promoted by IBM2, EDM2 and SUVH456 mitigates this instability. A very interesting and convincing result that I would emphasize more.

6) Typos:

- o Line 46: "promoters"
- o Line 143: "and TE-ALE TE-associated alternative 144 last exon (TE-ALE)"
- o Line 366: I think the authors meant fig7
- o Lines 800-801: *Capsella rubella*

9) Clarity:

- o What are "gene-like TEs"? Authors should explain the concept, a reference is not enough

(line 135)

- o For clarity, the diagram at the top of fig 1b should be referenced earlier, in the first paragraph of the results, to explicitly explain that TE-Gt are any transcript that contains a TE sequence.
- o The resolution of figures in the provided pdf is so low that the text is barely readable.
- o Fig 1b should contain the total number of transcripts, to reflect the proportion of transcripts that have a (detectable) TE sequence in the first place
- o Fig 1e should be referred to in the section entitled "ParasiTE: a tool for the detection of TE-Gts and their alternative isoforms"
- o Fig 2c: the scale must be different for CHG and CHH contexts (don't plot outliers), otherwise the distribution of the data is impossible to evaluate.
- o Unless I missed it, fig 4c is not referenced in the main text
- o about supplementary note 1: "ParasiTE detects gene annotations overlapping TEs by at least 80% in length". I assume this is 80% of the gene annotation, but it should be explicit.
- o References in the supp note 1 are missing or are wrong if they refer to the main text references

Responses to the reviewers' comments

First of all, we would like to thank the reviewers for their constructive and valuable comments on our manuscript. Below, we indicated the major changes made in the revised manuscript, and further provided point-by-point responses to the reviewers' comments.

Major changes and additional results

- As suggested by the reviewer #2 and #3, abbreviations were removed, redefined, or changed for better readability. In particular, "TE-gene transcript" and "alternative TE-gene isoform" were abbreviated as "TE-G transcript" and "ATE-G isoform", respectively, in the revised text and figures. In addition, as requested by reviewers, TE superfamily abbreviations such as "RLC" were removed from texts and figures.

- As suggested by the reviewer #3, we revised Figure 2, including the results of TE length in Fig. 2b, methylation of TEs in Fig. 2c, and nucleotide enrichments in Fig. 2d. Detailed explanations were provided in the response to the comment.

- As suggested by the reviewer #3, we further provided results regarding the enrichment of nucleotides in splicing acceptor/donor sites in TEs. We added results for all TEs, as well as each TE superfamily. We revised Fig. 2e and added/revised Supplementary Fig. 4 and 5. The "Materials and methods" section was also revised.

- As suggested by the reviewer #3, we have performed quantifications of alternative TE-Gene isoforms as well as of associated epigenome alterations.

We have provided additional data showing the magnitude of the isoform switching of each ATE-G isoform as new Supplementary Figure 7. There, a threshold of difference in isoform usage (dIF) ($|dIF| > 0.01$) with q -value < 0.05 was applied for the detection of significant isoform switching events. All the raw data of ATE-G isoforms were also updated and provided in Supplementary Data 5 and 6, and the "Materials and methods" section was revised.

- As suggested by the reviewer #3, we analyzed the correlation between differential isoform usage of ATE-G and chromatin change in mutants using Illumina Bisulfite-Seq data, and results were added as new Supplementary Figure 8, 9 and 10. We used our unpublished bisulfite-seq data (for *ibm1*, *ibm2*, *edm2*) and previously published data (for *suvh456*), and the related Materials and methods section was revised. All figures showing the percentage of DNA methylation comparison were re-plotted using only TEs containing at least 5 cytosines covered by at least three reads. This improved the accuracy of analysis, especially for DNA methylation calling from Illumina Bisulfite-Seq.

-As suggested by the reviewer #3, we have added new results about fractions of TE-ATTS and TE-ATSS in epigenetic mutants as Supplementary Figures 9a and 10a. In addition, we have added quantifications of epigenome changes associated with TE-ATTS and TE-ATSS events in the mutants as Supplementary Figure 9b, c, d and 10b, c, d.

In a response to the comment by reviewer #3, we added new results suggesting that the loci showing ATE-G isoform were epigenetically modulated by heat shock. We added analysis of transcriptome and methylome data under heat shock condition from a recent study (Nozawa et al., 2022). We revised Figure 7, Supplementary Figure 25 and 26, and Supplementary Table 4 with those data. Detailed explanations were provided in the response to the comment.

Additional changes

-Supplementary Figure 1b was re-plotted due to inappropriate scales in y-axis in the previous figure.

-During the revision, we noticed that Illumina RNAseq reads of *met1*, *ddm1*, *svh456* were duplicated during the data storage. We therefore reanalyzed the dataset and have revised the expression values in Supplementary Data 5 and 6, and re-plotted the related Figures: Fig. 3a and c, Figure 7 and Supplementary Fig. 25. This has not affected the conclusion of the manuscript.

-Supplementary Figure 2a (left plot) was re-plotted as we noticed some genes had been counted twice. This has not affected the conclusion of the manuscript.

-Supplementary Figure 19c was re-plotted to make it more informative.

-We corrected an error in Supplementary Note; “published gene-like TE annotation data by Panda et al. were concatenated.” and not “merged with bedtools’ merge function”.

-We added two references in the introduction: Duan et al 2017, Zhang et al 2021 and Zhang et al 2023, as well as other references suggested by reviewers as detailed below.

-We revised all qPCR bar graph to show data points according to the Nature Policy.

Point-by-point response to the reviewers' comments

Reviewer #1: In this manuscript, the author provided a global characterization of TE-gene transcripts (TE-Gts) in *Arabidopsis* by using Nanopore direct RNA-seq and the self-developed bioinformatics pipeline "ParasiTE". They identified the enrichment of TE superfamilies associated with different TE-Gt alternative isoforms. By analyzing the epigenetics modification, they found that the heterochromatic marks were required for Pol II transcription and impacted TE-ATTS (alternatively transcription termination sites) formation. Lastly, they investigated the environmental response to TE-Gts. Overall, this study provides a rich resource of chimeric gene-transposon transcripts in *Arabidopsis*. And combing various epigenetic mutant data, they illustrated the association between epigenetic state and TE-gene interaction, providing a new angle for studying transcription regulation. The comments below are intended to strengthen the study.

Major comments:

1. The authors detected a total of 11,348 TE-Gt and 6,025 altTE-Gi events. Did they consider the read coverage bias? Since the pore block and the 5'-3' degraded RNA transcript can result in truncated reads, leading to a 3' bias of ONT-DRS results (PMID: 32033565). Such bias may lead to an overestimation of isoforms.

Our response: As pointed out by the reviewer, 5'-3' degraded RNA transcripts may result in truncated DRS reads, leading to a bias of ONT-DRS coverage towards 3' ends of transcripts. We were aware that this issue could cause an underestimation of TE-ATSS events. To address the issue, we have included ONT-DRS data from Parker et al 2020, obtained by cap-dependent ligation of a biotinylated 5' adapter RNA to enrich capped mRNAs. In addition, we combined the latest AtRTD3 transcriptome dataset which consists of transcripts supported by high-confidence TSS/TES data, to obtain a non-redundant set of DRS-AtRTD3 transcript annotation. We think the DRS data from capped transcripts and AtRTD3 would have minimized the effects of truncated transcripts compared to DRS-only transcriptome analysis.

2. It is surprising that heterochromatic modification, such as H3K9 and DNA methylation, were required for Pol II elongation in some specific loci. From Fig.4a, the 3' terminal of long readthrough read seems coincident with PAS. Is it possible that this observation is associated with the choice of different PAS?

Our response: Indeed, our results demonstrated that the 3' terminal ends of long readthrough transcripts coincided with PAS. Although the mechanism of the choice of PolyA site is currently unclear, we speculate that changes in DNA and H3K9 methylation is associated with the choice of different PAS by RNA PolIII.

3. Fig3a and Fig3b showed that there are differences in the number of altTE-Gis among mutants based on the short-read RNA-seq datasets and DRS data. The author explained the smaller number of altTE-Gis due to the low coverage among other mutants. While the higher number of altTE-Gis occurs in *ddm1*, suggesting the dramatic impact of *ddm1*. So if the coverage of the data increases, is it possible to see changes in other mutants?

Our response: As commented by the reviewer, deeper sequencing of mRNAs would improve the detection of ATE-G isoforms in the mutants. On the other hand, the higher number of genes with ATE-G isoform detected in *ddm1* (Fig. 3a) compared to other mutants was likely due to its broader impacts on epigenome regulation, and we therefore speculate that the relative proportion of genes with ATE-G isoforms among mutants may not be largely affected by sequencing depth.

4. How the author explains or conjectures that in *met1*, the pol II enrichment increased (Fig. 4c). It is not mentioned in the paper.

Our response: We think the increase of PolII in TEs with ATTS might be a result of recruitment of PolII due to re-activation and transcription of TEs in *met1*. We mentioned this point in the revised main text.

5. The DRS raw data in fast5 format should be deposited as well.

Our response: DRS raw data and Fast5 data have been deposited in ENA under the accession number PRJEB53881.

Minor comments:

1. The DRS-*AtRTD3* annotation should be provided either as a supplementary file or via the website <https://plantepigenetics.oist.jp/>.

Our response: The DRS-*AtRTD3* annotation and other data sets are now available in our lab website <https://plantepigenetics.oist.jp/>.

2. Line 215, 'H3K9m' the description is not entirely correct. H3K9 methylation or the abbreviations 'H3K9M' is more appropriate. And in line 271, the word changes to 'H3K9'. Please contextualize.

Our response: We corrected H3K9m to H3K9 methylation throughout the manuscript.

3. Fig. 4b, the line color of different mutants was not annotated. It is difficult to correspond with each other.

Our response: This was caused due to an error during the file conversion. We corrected the color legend.

4. There seems to be something wrong with the correspondence of the figure to the article, such as in line 408, please check again.

Our response: We thank the comment. We corrected the reference to figures in the main text.

Reviewer #2: Berthelie and colleagues use ONT DRS in *Arabidopsis* mutants linked to epigenetic effects on the expression of TEs. The authors develop a computational pipeline focused on TEs. Since TEs are often repetitive in sequence ONT DRS with longer reads offers an advantage over short-read NGS methods. Specifically, the authors generate ONT DRS data for heavily used mutants in the field (*met1*, *ddm1*, *suv456*, *ibm1*, *ibm2*, *edm2*). These data are of good quality and the authors performed 2 repeats, but the read depth is rather low. The low read depth required the authors to rely on previous ONT DRS data to increase read counts for wild type that was already utilized for similar approaches. The authors identify a range of scenarios how RNA derived from TE sequences can interact with mRNA expression near the TE. The authors integrate analyses for some loci where the interactions of TEs with mRNAs have functional effects and were studied in detail previously. Overall, I believe that some of the data and analyses tools will be of interest to the TE community. The manuscript is very dense with acronyms, which makes it suitable for a specialized audience. The manuscript covers a lot of ground, which I appreciate since it is ambitious, yet here some conclusions need to be phrased more carefully as some conclusions have more the character of hypotheses. The authors also missed the windfall benefit of important studies in the field where some of the findings are already in the data. All in all, the manuscript offers some interesting aspects. The strength of the manuscript is that the associated data could be a useful as resource, but here I am unsure if the read depth will be sufficient to be taken up as such by the community.

Comments:

- The authors need to make the manuscript more readable. The authors use far too many acronyms. Many of the acronyms are not defined, or it is not presented when first mentioned. What does ParasiTE stands for? TE-IR, TE-ES, TE-Gts, altTE-Gi, TE-AS, TE-ATTS etc need to be defined at first mention. A new, high quality figure that clearly explains all the acronyms is warranted, but they need to reduce the acronym density.

Our response: We thank the comments for the improvement of the manuscript. “ParasiTE” is the name of newly developed pipeline in this study, and it was just a play on words about “TE” and “parasite” as TEs have been regarded as “molecular parasites” in genomes. We re-defined acronyms that were not defined before (TE-IR, TE-ES, TE-AS, TE-ATTS etc.) in the second paragraph of the result section. The associated RNA processing events were visually represented in figure 1c. In addition, “TE-Gene transcript”, and “alternative TE-Gene isoform” were abbreviated as “TE-G transcript” and “ATE-G isoform”, respectively, in the revised text and figures, as suggested by the reviewer #3. We also edited the first 4 paragraphs of the result section to be concise and more readable, by removing the numbers and descriptions related to methods. We also removed abbreviations that were not frequently used in the text. The compressing and PDF conversion had generated low quality figures during the initial submission, and we have provided high quality figures separately in addition to the all-in-one PDF, which we hope are readable in the revision.

By looking at the examples in this MS and data from other sources I got the impression that at least a fair fraction of the transcripts that the authors find in chromatin mutants are in fact already detectable in nuclear RNA degradation mutants. I find this is an important conceptual point the authors have missed, it challenges the hypothesis that the transcript are “only” detectable in the chromatin mutants. Yes, expression is probably induced as expected and often already shown, but many isoforms can be detected as “cryptic RNAs” without disrupted chromatin repression (PMID: 43244691, 31863587).

Our response: One of the main findings of this study is the identification of continuous TE-gene chimeric transcripts and detection of RNA processing events generating alternative TE-gene transcript isoforms in the reference transcript annotation of *A. thaliana* as well as in transcriptome of chromatin mutants. We agree that these TE-G transcripts may also be regulated in particular conditions or mutants other than the chromatin mutant backgrounds, and indeed another important finding of this study is that various environmental conditions (including heat shock) can regulate ATE-G isoform production and isoform switching, similarly to the chromatin mutants (Fig. 7).

There are contradictions in the presentation of novelty: “limitations of short-read RNA sequencing and a lack of bioinformatic pipelines have hindered the detection and comprehensive analyses of these transcripts in plants.” Lines 91-92. Then: “We found that about 3,000 Arabidopsis genes are associated with TE-Gt production, corresponding to about 8% of protein-coding genes annotated in AtRTD3 (Fig. 1), which is close to an estimation made by a previous study.” Lines 383-385. If they find the same number of these transcripts, there surely must have been already a method to identify them.

Our response: Lockton et al 2009 (<https://doi.org/10.1007/s00239-008-9190-5>) conducted a BLAST similarity search between a set of genes and TEs, which was different from our approach. In this study, we obtained DRS dataset and developed the ParasiTE pipeline, which allowed us to specify the RNA processing events producing TE-G transcripts for the first time. In addition, we have identified novel ATE-G isoforms and their switching events associated with epigenome changes in the mutants as well as environmental stresses, which have been experimentally verified including stress treatments. Thus, we have addressed

several important biological questions in this study that have not been explored before, including epigenome and environmental regulations of TE-G transcripts. In addition, our study provides a novel bioinformatic pipeline and new public transcriptome and epigenome datasets, which would be original novel resources for future studies in the scientific community.

- How were the two ONT DRS repeats treated? Like true repeats, as in – the tracks only show RNA isoforms consistent in both repeats – or, where the repeats just merged? In case the repeats were merged without the removal of reads that support isoforms by some singleton jackpot reads specific to one of the repeats, the benefit of the two repeats is somewhat diminished. The authors should be clear about how they treated the repeats in the main text. I got the impression that the low depth of their ONT DRS data may have forced the authors to merge the reads of their repeats (and use prior data for wt).

Our response: In this study, Col-0 DRS reads were obtained from two biological replicates, which were combined with DRS data of Col-0 from a previous study (Parker et al 2020 <https://doi.org/10.7554/eLife.49658>). Next, “DRS-Col-0” transcriptome annotation was generated using Stringtie2 and merged with either Araport11 or AtRTD3 reference using Stringtie2 merge function, obtaining the DRS-AtRTD3 or DRS-Araport11 reference transcriptome dataset. ONT-DRS is a relatively new technology, and each study is still trying to find the best practice of transcriptome assembly. Our approach can increase the transcriptome diversity by combining additional DRS datasets. Actually, a similar methodology was applied for Arabidopsis DRS analysis in Parker et al 2021. (<https://doi.org/10.7554/eLife.65537>) ; They wrote: “Transcriptional loci were first identified in Col-0, fpa-8 and 35S::FPA:YFP Nanopore DRS reads using the long-read transcript assembly tool StringTie2 version 2.1.1 (Pertea et al., 2015). Novel transcriptional loci were merged with annotated loci from the Araport11 reference (Cheng et al., 2017).” For the readability and according to the comment by the reviewer #3, detailed explanations of the methodology was described in the method section.

- it would have been good to see if WT has been included in the Fig. 3a to compare the isoforms in WT to the mutants.

Our response: As suggested, we added WT Col-0 data to the Fig. 3a.

- How big is the overlap of their transcript isoform detection method and other approaches? For example data-driven transcriptome reconstruction (PMID: 34058980) and alternative RNA isoform detection methods (PMID: 32444691)?

Our response: As requested by the reviewer, we compared the overlap of transcripts of DRS-AtRTD3 transcriptome against the transcripts of “PMID_34058980” or “PMID:32444691” transcriptomes. For the analysis, we converted bed files in gtf using “BED-to-GFF” from “<https://usegalaxy.org/>” and added unique ids for each of transcript when necessary. Next, we used “bedtools intersect -f X -r”, with X reciprocal overlap fraction of 100, 98, 95 or 90% required for detection of overlap between transcripts of DRS-AtRTD3 and transcripts of each compared transcriptome. Results of the percentage of overlapped transcripts are shown below:

	Total transcripts	Overlap 100%	Overlap 98%	Overlap 95 %	Overlap 90%
DRS-AtRTD3	199,489	/	/	/	/
PMID_34058980 (Called_transcripts.bed)	72,870	813	28,936	46,155	56,979
TIF-seq_PMIID_32444691 (col-0.allreads.bed)	294,943	62	74,635	142,246	192,497

- It would have been good to see if there was any comparison made between this tool and existing tools to show the advantages of ParasiTE over them.

Our response: To our knowledge, no tools comparable to ParasiTE is available to date. ParasiTE can detect a wide range of ATE-G isoforms (TE-IR, TE-ES, TE-A3SS, TE-A3SS, TE-ATTS, TE-ATSS, TE-ALE, and TE-AFE), by using a transcriptome annotation.

In parallel to our initial submission of the manuscript, a preprint of a new pipeline called ChimeraTE was online in Biorxiv (<https://doi.org/10.1101/2022.09.05.505575>), which may be urged by our study. However, this tool does not identify the same range of ATE-G isoforms as ParasiTE and has been developed for paired-end short RNAseq data as input.

- The ATSS part can be enhanced by providing context, key publications that address the roles of chromatin on the regulation of intragenic TSSs (PMID: 30707695, PMID: 32863101).

Our response: As suggested, we mentioned the context about the regulation of intragenic TSSs in the TE-ATSS result part and cited the proposed publications in the revised manuscript.

- CAGE data and ONT should be scaled, normalized (page 53, b)

Our response: We thank the reviewer for the comment. We added a normalized scale information.

- Fig2b-at2g15790-transcripts are already visible in plaNET-seq data (PMID: 31863587). Fig2c should be at2g40955.

Our response:

We think the reviewer is mentioning Fig. 3b and c (not Fig. 2b, c). The isoform of at2g15790 has been identified in Yan, X. et al. 2016 and Tu et al. 2020 as referenced in the manuscript, and we also cited the study of plaNET-seq (PMID: 31863587) in the revised manuscript. Regarding Fig. 3c, we thank the reviewer for the comment and we corrected as AT2G40955.

- Fig4a, at4g16860 and at5g13200 (also Fig. 5a) already in TIF-seq data (PMID: 32444691), especially hen2 mutant TIF-seq data. Fig4a, at5g13200 even already called as MC transcript (PMID: 34058980).

Our response: As the reviewer commented, we found that a subset of TE-gene transcripts of at4g16860 and at5g13200 were present in previous transcriptome annotation dataset of *A. thaliana* (PMID: 32444691 and 34058980). The studies were cited in the revised manuscript. However, no information about TE-Gene chimeric transcript were provided in these datasets and related publications. In addition, as shown in the tracks below, we found that the DRS-AtRTD3 transcriptome in our study showed better resolutions in terms of the transcript diversity and the number of ATE-G isoforms with RNA processing (splicing)

information. In addition, we also provided various epigenome data and stress conditions associated with the TE-G transcript annotations.

- citations 68, uses the author 1st name instead of last name.

Our response: We thank the reviewer for the comment. We corrected the citation.

- How are the authors propagating the chromatin mutants? Are they regenerating heterozygotes so that we can be reasonable confident that we are looking at real effects of the epigenetic mutant rather than composite effects of increasingly deteriorating epigenetic indirect effects and the direct epigenetic effect in these mutants? A degen strain for rapid protein depletion of the factors would be a much better experimental approach.

Our response: We maintained *ddm1*, *met1*, *ibm1*, *ibm2*, *edm2* mutations as heterozygous for propagation, and all mutant plants in the second generations of homozygous were used for the DRS, mRNAseq, BS-seq, and ONT-DNA methylation analyses. In addition, all mutants and wild-type were harvested at the 10-day-old stage which would minimize generation effects. *svh456* was maintained as triple mutants while it is indistinguishable to wild-type Col-0 in terms of the morphological phenotype. We are not aware of an established degen system practically available in *Arabidopsis*. Auxin-inducible degen (AID) in animal and yeast systems are not compatible to plants and essentially a tangent assay system suitable for cell cultures.

Reviewer #3: Key results: Bertheliet al. developed a pipeline that leverages long-read transcriptomics to evaluate the contribution of transposable element (TE) sequences to the diversity of gene transcript isoforms. ONT-DRS data from multiple mutants show how gene-TE transcript genesis depend on maintenance of heterochromatic marks and on IBM2 and EDM2, factors required for heterochromatin transcription. Importantly, the authors highlight how selected alternative gene transcripts arise from recent TE insertions which ultimately influence mRNA isoform stability and gene responsiveness to environmental cues.

Significance: The study is highly original, provides important insights in deciphering the role of TE polymorphisms into environmental cellular response and potentially, local adaptation. The authors made the effort of designing a bioinformatic pipeline adaptable to other datasets and

organisms, which is a significant contribution to the scientific community.

Data & methodology: Experiments showing the physiological relevance of the TE-gene transcripts were carefully controlled and conducted. Data analysis and representation has been well conducted, with some exceptions detailed below. The supplementary note 1 explains well the authors methods and strategic choices for the development of their design, showing a thoughtful design. Statistical tests are appropriately used.

Validity: The study suffers from minor flaws that should be addressed but do not affect the overall conclusions of the study. Most importantly, the manuscript's exciting biological insights are concealed by a lack of conciseness and an overuse of jargon, numbers and abbreviations in the result section that will discourage the reader, especially in the results section. This is particularly striking as the discussion is well written. The authors should try to clarify their message and improve the writing of the result section.

I recommend the editor to invite the authors to revise their manuscript before acceptance.

Major points:

1) Some of the reported phenomena are not quantified or described sufficiently

Figures 4, 5 and 6 and related supplementary figures are mostly based on genome browser snapshots that provide convincing examples, but the authors did not quantify some of these observations even though they have the data in hands.

Our response: As suggested, we reanalyzed transcriptome and epigenome data and added new results about degree of ATE-G isoform switching, epigenome changes in TEs associated with ATE-G isoform, and in TEs especially associated with ATSS and ATTS. Detailed explanations were provided in each response below.

Magnitude (line 213): “10–20 % of altTE-Gis showed significant differential isoform usage”. This is a key result that the authors should describe in more details. What is the magnitude of the effect? Authors could show the distribution of isoform fold changes and inform the reader about the threshold used here for significance.

Our response: As suggested, we have provided additional data showing the magnitude of the isoform switching of each ATE-G isoform in each mutant as new Supplementary Figure 7. A threshold of difference in isoform usage (dIF) ($|dIF| > 0.01$) with isoform switch q-value < 0.05 was applied for the detection of significant isoform switching events. All the raw data of ATE-G isoforms were also provided in Supplementary Data 5 and 6.

Correlation with changes in mDNA & H3K9me2: in “Alteration of epigenetic modifications impacts TE-Gt production”, authors analyzed chromatin mark changes only for selected gene snapshots. They should plot changes in heterochromatic marks for all TEs associated with differential isoform usage in the mutants. This would help determine if isoform switching is a direct consequence of the loss of heterochromatic marks or not.

Our response: As suggested, we reanalyzed epigenome changes of TEs associated with overall ATE-G isoforms and new results were added as Supplementary Figure 8. Significant changes in DNA methylation of TEs associated with ATE-G isoforms were detected in *met1* and *ddm1* compared to Col-0 (Supplementary Figure 8a). For a consistency, changes in methylation at CHG context was quantified as a proxy of H3K9 methylation in TEs associated with ATE-G isoforms detected in *ibm1*, *suvh456*, *ibm2*, and *edm2*. We found a significant decrease of CHGm in *suvh456* in TEs associated with ATE-G isoforms, indicating that differential usage of ATE-G isoforms correlates with a change in heterochromatic marks as concluded (Supplementary Figure 8b).

Regarding *ibm1*, the average of CHGm in TEs associated with ATE-G is slightly higher in *ibm1* compared to Col-0, but the difference was not statistically significant (Supplementary Figure 8b). *ibm1* is known to show an increase of H3K9me2/CHGm at gene bodies, and

therefore we analysed CHGm at the TE-gene locus (comprising gene and TE region involved in ATE-G isoform production) and found a significant increase in *ibm1* compared to Col-0, consistent with previous reports. Currently, the cause of the isoform switching of ATE-G found in *ibm1* remains to be explored.

In *ibm2* and *edm2*, no statistically significant changes in CHGm were observed at TEs or TE-gene locus (Supplementary Figure 8b and c), consistent with previous reports (Saze et al. 2013, Le et al. 2015). The isoform switching of ATE-G isoforms found in *ibm2* and *edm2* were mainly caused by the dysfunction of the IBM2-EDM2 complex that is thought to act downstream of H3K9me2. We added description of the results in the revised text.

Quantification: in “Epigenetic regulation of TE-ATTS formation in A. thaliana”, changes in TE-ATTS in the mutants have not been quantified by the authors. This is needed to give a reader a sense of how much TE-ATTS contributes to the 10-20% of alTE-Gis with significant isoform switching in the mutants. Similarly, among TE-ATTS events, how often the mutants show premature termination, and how often do they show early termination?

Our response: As suggested, we added new results about fractions of TE-ATTS and TE-ATSS in epigenetic mutants as new Supplementary Figure 9 and 10. Overall, genes with TE-ATSS and TE-ATTS each represent about ~40-50% of genes associated with all ATE-G isoform production events in each mutant (Supplementary Fig. 9a and 10a), representing the major fraction of ATE-G isoform with change in isoform usage.

It was not clear for us the difference between premature termination and early termination, and in our analysis all ATTS overlapping with TE sequence were counted as TE-ATTS events by ParasITE.

How does that correlate with changes in epigenetic marks? Most of this paragraph is based on genome browser snapshots, with convincing examples, but the reported phenomena must be quantified. Same observation for TE-ATSS. One could guess from fig7 and supp fig19 that these numbers are small, but if that is the case, authors should explicitly report them and comment.

Our response: We quantified changes in epigenetic marks associated with TE-ATSS and TE-ATTS as suggested, and added new data as Supplementary Fig. 9b, c and 10b, c. We found a significant decrease in CG/CHG methylation in TEs associated to TE-ATSS and TE-ATTS in *met1* and *ddm1* (Supplementary Fig. 9b, c and 10b, c).

We also analyzed CHGm as a proxy of H3K9 methylation in *svh456* and *ibm1* mutants as above, and found a significant decrease in CHGm in TEs associated with TE-ATSS and TE-ATTS in *svh456*, especially in TE-ATTS (Supplementary Fig. 9d and 10d).

2) Overinterpretation

- “DRS of the Arabidopsis transcriptome”

Line 172-173: “we found that TE sequences associated with TE-Gts or altTE-Gis were mainly located in 3'-UTRs of genes”. This conclusion is not supported by the data. Supp fig2b shows that TE-Gt & altTE-Gi are more frequent in 3'-UTR than CDS & 5'-UTR, but the difference is small and the phrasing should reflect this. The current phrasing is misleading.

Our response: We agree with the point. We revised the text as “we found that many TE sequences were associated within the first and last exon of TE-G transcripts (Supplementary Fig. 2a), overlapping with 5'/3'-UTR of gene transcripts (Supplementary Fig. 2b).”.

- “Contribution of TE superfamilies to TE-Gt production”

This part of the results is not very convincing. The authors report some minor over and under-representation of TE superfamilies and families. The fold change of observed / expected are not striking (fig 2a, 2d). I would rather conclude that all superfamilies contribute to TE-Gt, TE-AS

and TE-ATP, with some superfamilies / families contributing more to the phenomenon. For example, the fold change enrichment for TE families in fig 2e) is of 1.63 for ATIS112A, 2.70 for ATENSPMI + ATENSPMIA (supp data 3). Those are not very high numbers. The ATMU6N1 family shows the strongest enrichment (13.31, supp data 3), why not show the splicing donor / acceptor sites for this family?

Our response: As suggested, we further analyzed splicing sites overlapping of ATMU6N1 family instead of the Vandal17 which has the higher enrichment in TIR/Mutator Superfamily, and the results were added in Supplementary Fig. 5. Although statistically significant, we agree that the enrichments were not so remarkable. We revised the text about contribution of particular TE superfamilies to the ATE-G isoform productions in the paragraph. To be more general about the role of TE sequences in TE-G transcripts, the conclusion was revised as “Overall, these results suggest that TE sequences in intragenic regions can be incorporated into TE-G transcripts by providing splicing donor/acceptor sites.”.

More importantly, a canonical splicing donor site is GU, while an acceptor site is AG. Did the authors observe different consensus sequences with other TE families? Or when they merged all families together? If that is the case, then they could indeed conclude that “Overall, these results suggest that enrichment of particular TE superfamilies in intragenic regions contributes to splicing events of a subset of genes.”. In the current state, there is no evidence to support this conclusion.

Our response: As suggested, we further analyzed enrichment of nucleotides in splicing acceptor/donor sites for all TEs, as well as each superfamily, and added as Figure 2e and Supplementary Figure 4. We also revised our methodology and results. We found that TE sequences associated with splicing acceptor/donor were enriched with GT-AG nucleotides in all TE family, suggesting that splicing machinery can recognize the motifs regardless of the TE superfamily.

What can be concluded from fig 2b? This plot likely reflects the difference in size of transposon superfamilies themselves. One would have to look at the length of TEs associated to altTE-Gi after normalizing to the superfamily average size. The plot in its current state does not provide any valuable information. Therefore, the sentence “Most of the TE sequences associated with altTE-Gis were short and truncated, while retrotransposons (Copia [RLC], Gypsy [RLG], and LINE [RIX]) superfamily sequences were relatively long when compared with other TE superfamilies (Fig. 2b).” (lines 183-185) should be removed / reworked. The proper way to look at TE size would be to ignore TE superfamilies, or to plot the data as the authors did on fig S3c.

Our response: As suggested, we reanalyzed TE sizes associated with ATE-G isoforms for all TE families and replaced to Fig. 2b. The result showed that TEs associated with ATE-G isoforms are generally shorter than intergenic TEs, while still longer than intronic TEs, suggesting that these TEs may be relatively young and less degenerated. Indeed, they are often polymorphic among *Arabidopsis* accessions (Fig. 5d).

- “Environmental stresses affect alternative transcription termination by TE sequences”

“These data suggest that environmental signals can regulate the transcription and processing of altTE-Gi via epigenetic regulation, which may contribute to an adaptive response to environmental changes.” The authors did not provide a single piece of evidence that the loci showing altTE-Gi were epigenetically modulated by environmental stresses. Stresses are known to affect splicing patterns, and there is no evidence to my knowledge that there is an epigenetic basis to this phenomenon. If authors really want to make this point, they will at least have to show that alternative isoform usage at TE-Gt upon stresses correlate with changes in chromatin marks. The data is still very interesting as such, showing that TE insertions can affect transcript isoform stoichiometry in response to environmental perturbations.

Our response: We agree with the comment by the reviewer. In a response to the comment, we further analyzed transcriptome and methylome data under a heat stress condition obtained from a recently published study (Nozawa et al., 2022), and added as new results in supplementary Fig. 26 (revised version of the previous Supplementary Fig. 20). The results demonstrated that at least the two loci containing the heat responsive *COPIA78/ONSEN* showed a reduced DNA methylation in TE regions (although the coverage of cytosine in *ATITE11295/ONSEN* is low), which is coincided with isoform switching events. It is still a correlation between the isoform switching and epigenome changes induced by a stress and this may be specific to the loci containing the heat-responsive *ONSEN*. Nevertheless, the loss of the long ATE-G isoforms associated with a loss of DNA methylation is consistent with our observations in other loci (such as loss of ATE-G isoforms of RPP4-AtCopia4 in *svvh456*; Fig. 4) where heterochromatic marks enhance transcription of TE-G transcripts. We revised the text in the manuscript.

Minor points:

1) Rework the first four subsections of the results section (lines 106 to 204)

These four subsections are unappealing and should be reworked.

- Only methods that are relevant to data interpretation should be explained in the results, other details should be moved to the Methods section or if necessary, to the figure legends.

Our response: As suggested, the first four paragraphs in the result section were revised to be concise and to improve readability. Descriptions related to the methodology were moved to the method or removed. Specific changes were detailed in the following responses.

- The authors constantly refer to DRS-AtRTD3 and DRS-Araport11. The authors made it clear that DRS-AtRTD3 was their reference of choice in the 1st subsection, so they can spare this detail. Whether any number comes from an analysis with one reference or the other belongs to the figure labels or the figure legends.

Our response: As suggested, we removed sentences referring to DRS-Araport11 in the revised text, while the data were referred in the figure legends and methods.

- About "Identification and classification of TE-Gts in A. thaliana": this subsection has too many numbers to be readable. All numbers are already shown in figure 1, percentages are enough for the text and will improve readability.

Our response: As suggested, only percentages of TEs were indicated in the revised manuscript.

- What information provides the following sentence (line 156): "27,628 genes were associated with Arabidopsis Genome Initiative [AGI] codes"? Remove or explain to the reader why this is relevant.

Our response: Most of the AtRTD3 transcripts are annotated with the corresponding Arabidopsis gene code (such as ATxGxxxxx) (27,628/39,998 gene models), while remaining transcripts were not associated to gene ID. This sentence is moved from the legend of Figure 1, without using the abbreviation.

Abbreviations should be removed as much as possible and reworked. The diversity of splicing events requires the extensive use of abbreviations, therefore an effort should be made to avoid other abbreviations, especially when they are used only once. I suggest removing: AREs (used once in the text), CTD (once), BP (once in fig1), CC (once in fig1), MF (once in fig1), AGI (three times), dIF (once in fig7), possibly others that I missed. Abbreviations for TE superfamilies seem only relevant to give more space to figures 2b and 2d. I suggest getting of them entirely and reworking the figure. A horizontal barplot would give more space to the figure for example.

More importantly, abbreviations should be more intuitive to interpret:

- “Alternative” is sometimes abbreviated with “A” (APA) and sometimes with “alt” (altTE-Gis), this is confusing.

- The abbreviations combining consecutive uppercase and lowercase letters to distinguish consecutive words are very unintuitive, making the reading quite unpleasant. Here are my propositions for improvement:

o TE-Gts: TE-G transcripts or TE-G-Ts

o altTE-Gis: ATE-G isoforms or A-TE-G-Is

o Epi-altTE-Gi: Epi-A-TE-G isoforms

Our response: We thank the comment for improvement of readability of the manuscript. As suggested, we removed abbreviations appeared less frequently as following:

-ARE; it was removed in the main text, while it is used to indicate the motifs in Supplementary Fig. 20 and 24.

-CTD; it was used once in the main text, twice in methods, and once in the legend of figure 4. Therefore, we kept the abbreviation.

-BB, CC, MF, AGI; removed.

-dIF; it was used in Fig. 7, Supplementary Fig. 6 new Supplementary Fig. 7-10, Supplementary Fig. 25, and we therefore kept the abbreviation.

-Abbreviations for TE superfamilies; removed from the text and figures.

-APA, TE-Gt, altTE-Gis; As suggested, we changed “TE-Gt” to “TE-G transcript”, “altTE-Gi” to “ATE-G isoform”, and “Epi-altTE-Gi” to “Epi-ATE-G isoform” throughout the manuscript and figures. APA was kept as it was used more than 20 times in the manuscript.

-TE superfamily abbreviations (such as “RLC”) were also removed in the text and figures.

2) *Does the figure of 39,998 gene models include lncRNA genes, ribosomal RNA genes, transfer-RNA genes, micro-RNA genes, etc.? Or is it only protein-coding genes? This information should be provided to the reader (I didn't find it in the methods).*

Our response: As commented, the 39,998 gene models of DRS-AtRTD3 include transcripts in Chr1 to 5 of AtRTD3, therefore it also contains lncRNA genes, ribosomal RNA genes, transfer-RNA genes, micro-RNA genes in addition to the protein-coding genes. We provided this information in the method section.

3) *Why does the data aggregation of DRS-AtRTD3 detect 20% more genes with TE-Gts than Araport11 and AtRTD3? Could it be that overall transcript coverage is increased because more tissue types are involved when they combine datasets? Or is it simply because the authors sequenced deeper?*

Our response: DRS-AtRTD3 could detect more TE-G transcripts compared to Araport11 and AtRTD3, benefitted from DRS long reads. We found a higher number of TEs involved in TE-IR, TE-ATSS, and TE-ATTS in DRS-AtRTD3 which were not annotated in AtRTD3 or Araport11 data (Supplemental Fig. 1c). For instance, ONT DRS detected novel gene isoforms containing intragenic TE sequences, which were likely ignored or discarded in AtRTD3 or Araport11 (as Fig. 1a, or as our figure shown below).

4) *Supp fig3b: how does BS-seq detect DNA methylation in such a repeated region? If this region is repeated and unmappable uniquely by short-read sequencing, the only way to measure DNA methylation there would be to include multi-mapping reads, is that the case here? If not, there should be no signal at all in BS-seq data. Below is the same region with uniquely-mapped reads from BSseq data:*

Our response: We noticed that they were mislabeled in Supplementary Fig. 3b in the initial version of manuscript. Upper three tracks with higher coverage of cytosines were data from ONT DNA methylation calling and the lower three tracks were from BSseq. Only uniquely-mapped reads from BSseq data using Bismark were used for methylation calling.

5) In “Epigenetic regulation of TE-ATTS formation in *A. thaliana*”: the presence of the TE insertion induces instability (the CDS transcript is less upregulated upon ABA treatment, fig 5d), and the distal APA promoted by IBM2, EDM2 and SUVH456 mitigates this instability. A very interesting and convincing result that I would emphasize more.

Our response: We agree that the point is one of our key results and we have emphasized it more in the discussion:

“The instability seems to be mitigated in Col-0 under the normal condition by the production of the long ATE-G isoforms (Fig. 5d and e), while it became severe in *suvh456* and *ibm2/edm2* (Fig. 5e). This suggests a potential role of the epigenetic machinery to minimize the impact of intragenic TE insertions on the host gene expression.”

6) Typos:

o Line 46: “promoters“

o Line 143: “and TE-ALE TE-associated alternative 144 last exon (TE-ALE)”

o Line 366: I think the authors meant fig7

o Lines 800-801: *Capsella rubella*

Our response: We corrected the typos in the main text.

9) Clarity:

o What are “gene-like TEs”? Authors should explain the concept, a reference is not enough (line 135)

Our response: As suggested, we added details to explain the concept of gene-like TEs: “gene-like TE annotation and associated transcripts (annotation of TEs and associated transcripts included as genes in the gene annotation)” in the method section.

o For clarity, the diagram at the top of fig 1b should be referenced earlier, in the first paragraph of the results, to explicitly explain that TE-Gt are any transcript that contains a TE sequence.

Our response: As suggested, Fig. 1b was referred in the first paragraph of the result section, explaining TE-G transcripts and alternative TE-G (ATE-G) transcripts.

o The resolution of figures in the provided pdf is so low that the text is barely readable.

Our response: The compressing and PDF conversion have generated low quality figures during the submission. We have also provided high quality figures separately in addition to the all-in-one PDF, and we hope they are readable in the revision.

o Fig 1b should contain the total number of transcripts, to reflect the proportion of transcripts that have a (detectable) TE sequence in the first place

Our response: As suggested, we added the total number of transcripts (199,489) in the Figure 1b.

o Fig 1e should be referred to in the section entitled “ParasiTE: a tool for the detection of TE-Gts and their alternative isoforms”

Our response: As suggested, Fig. 1e (now Fig. 1c in the revision) was referred in the section.

o Fig 2c: the scale must be different for CHG and CHH contexts (don't plot outliers), otherwise the distribution of the data is impossible to evaluate.

Our response: As suggested, box plots of CHG and CHH contexts in Fig 2c were revised with an adjusted scale of the y-axis.

o Unless I missed it, fig 4c is not referenced in the main text

Our response: Indeed, it was not mentioned by a mistake in the initial manuscript. We referred Fig. 4c in the revised main text.

o about supplementary note 1: “ParasiTE detects gene annotations overlapping TEs by at least 80% in length”. I assume this is 80% of the gene annotation, but it should be explicit.

Our response: We agree with the reviewer, and we modified to make it explicitly: “ParasiTE detects gene annotations overlapped by TEs by at least 80% in length of the gene annotation.”

o References in the supp note 1 are missing or are wrong if they refer to the main text references

Our response: We are thankful to the reviewer for the comment. We added the references in the Supplementary Note 1.

REVIEWER COMMENTS

Reviewer #1 (Remarks to the Author):

The authors have successfully addressed all of my concerns.

Reviewer #2 (Remarks to the Author):

The response to the reviewers highlights major changes during the revision. The authors note that 7/7 of those address comments of reviewer #3. My original concerns were partially addressed. I do not intend to bring up new issues when re-revising a manuscript, but my original comments highlighted serious flaws of the manuscript that remain to be fully addressed. I encourage the authors to engage with the original comments on a higher level and rephrase some of them below.

Remaining major issues:

- It remains open to what extent this MS makes novel vs confirmatory findings. The overlap to prior data driven annotations that used Arabidopsis DRS data supplemented by other methods to rule out artifacts (PMID: 34058980) needs to be worked out much more clearly. It remains missing in the manuscript as a figure with appropriate space in the revised text. The transcript boundaries in RTD3 and Araport11 tend to be inaccurate, this is community knowledge and in case the authors were unaware it is also clear from PMID: 34058980. It is a conceptual concern if the transcripts are also present in prior studies examining different conditions. The table inserted in the reviewer response does not address my point. The exercise for the revision had been to truly engage with the high-quality transcriptome annotation of Ivanov et al, ideally match that quality, and convince that that the authors are largely adding new information, rather than concealing a rediscovery of what is already known and giving it a new name. I had the impression that there is a big overlap based on the detailed comments on loci were the authors presented genome browser views— many of them were not specific to the mutants. An overlap of 56,979/72,870 as indicated for 90% indicates that there is a large overlap between high confidence transcripts. The task the authors were given in the review process was to thoroughly address that systematically.
- I remain unconvinced about how the repeats of the ONT data were used. While repeats were performed, they were merged without harnessing the benefits of generating repeats. For example by implementing criteria to call novel transcripts only if present in several repeats etc to reach a quality threshold. This strategy deprives the authors of the opportunity to carefully perform QC on the number of reads from independent experiments that support novel transcripts. Using a prior ONT DRS set is somewhat OK, but it is of course much better to generate their own ONT DRS Col data of sufficient depth so that the experimental conditions are identical to those used to grow the mutants. How can the authors confidently rule out that a portion of the transcript isoform differences between the mutants and the highly merged Col dataset that is largely contributing to the control set is actually rooted in differences in growth conditions?

Remaining issues:

- The authors claim that ONT DRS is a new method. That was true a few years ago, but by now it is already around for many years. Issues such as the abortion of reads, resulting in the impression of artifacts misleadingly indicating 5'-isoforms are known issues. Next-level analyses are already published, for example done in PMID: 34058980. Simply using ONT DRS without addressing these biases is actually a step backwards from the knowledge foundation in 2022/23 rather than an advance.
- The authors worked on the readability regarding the previous acronyms to some extent. The benefit of using the author-defined acronyms over community standards remains unclear. Chimeric transcript have been studied and so there is no real need to reinvent the

wheel.

- The information on propagation of chromatin mutants remains missing from the methods.
- The authors should clarify the '-R' option of in the in the StringTie2 manual. A comparison of -L and -R modes and a valid, scientific comparison of both modes should be presented.

Reviewer #3 (Remarks to the Author):

The Authors addressed carefully my comments and I found just two remaining minor points to consider.

1. legends for the supplementary figures appear to be missing.

2. "As commented, the 39,998 gene models of DRS-AtRTD3 include transcripts in Chr1 to 5 of AtRTD3, therefore it also contains lncRNA genes, ribosomal RNA genes, transfer-RNA genes, micro-RNA genes in addition to the protein-coding genes. We provided this information in the method section."

I find it curious to represent the percentage of gene isoforms relative to the total number of genes (fig 1d) with the inclusion of lncRNA genes, ribosomal RNA genes, transfer-RNA genes, micro-RNA genes which are not spliced. This tends to exaggerate the perception that the phenomena described by the authors in this paper is only marginal. Could the Authors redo the calculation with only RNAs that may be prone to splicing?

Responses to the Reviewer's comments

We would like to thank the Reviewers for their comments on our revised manuscript. Below, we provided point-by-point responses to the Reviewers' comments to the revision. All the changed texts are reflected in the **blue color** in the (second) revised manuscript.

Point-by-point response to the Reviewers' comments

Reviewer #1 (Remarks to the Author): *The authors have successfully addressed all of my concerns.*

Reviewer #2 (Remarks to the Author):

The response to the Reviewers highlights major changes during the revision. The authors note that 7/7 of those address comments of Reviewer #3. My original concerns were partially addressed. I do not intend to bring up new issues when re-revising a manuscript, but my original comments highlighted serious flaws of the manuscript that remain to be fully addressed. I encourage the authors to engage with the original comments on a higher level and rephrase some of them below.

Remaining major issues:

- It remains open to what extent this MS makes novel vs confirmatory findings.

Our response: We respectfully disagree with this Reviewer's comment. Based on the comments, we believe there were misunderstandings by the Reviewer about the design, scientific focus, and scope of the study. We explain and clarify these points in the following responses.

We replied to this comment in our first revision as; *“In this study, we obtained DRS dataset and developed the ParasiTE pipeline, which allowed us to specify the RNA processing events producing TE-G transcripts for the first time. In addition, we have identified novel ATE-G isoforms and their switching events associated with epigenome changes in the mutants as well as environmental stresses, which have been experimentally verified including stress treatments. Thus, we have addressed several important biological questions in this study that have not been explored before, including epigenome and environmental regulations of TE-G transcripts. In addition, our study provides a novel bioinformatic pipeline and new public transcriptome and epigenome datasets, which would be original novel resources for future studies in the scientific community.”.*

We feel it was unfair that the points mentioned in our revision comment were not addressed properly or intentionally ignored by the Reviewer. In particular, the data presented for the epigenetic and environmental regulation of differential isoform usages in TE-gene transcripts, and examination of their biological significance (Main figures Fig3-7, Sup. Fig. 4-26) seemed to be overlooked by the Reviewer, according to the comments (see also below). Although the Reviewer kept mentioning the presence of similar transcripts in previous studies, many of TE-gene chimeric transcripts are novel or explicitly identified and with much higher resolutions and annotation, and experimentally validated for the first time in this study. Additionally, the studies cited by the Reviewer during the first revision comments never addressed the differential expression and isoform usage of TE-gene transcripts under various mutant backgrounds and environmental conditions, together with experimental verifications of the impact on plant environmental responses. In fact, these points highlighting the novelty and significance of this work were also echoed in the comments by Reviewer#1 and #3 below:

Reviewer #1 synthesis: “. Overall, this study provides a rich resource of chimeric gene-transposon transcripts in Arabidopsis. And combing various epigenetic mutant data, they illustrated the association between epigenetic state and TE-gene interaction, providing a new angle for studying transcription regulation.”

Reviewer #3 synthesis: “The study is highly original, provides important insights in deciphering the role of TE polymorphisms into environmental cellular response and potentially, local adaptation. The authors made the effort of designing a bioinformatic pipeline adaptable to other datasets and organisms, which is a significant contribution to the scientific community.”

“Importantly, the authors highlight how selected alternative gene transcripts arise from recent TE insertions which ultimately influence mRNA isoform stability and gene responsiveness to environmental cues.”

-The overlap to prior data driven annotations that used Arabidopsis DRS data supplemented by other methods to rule out artifacts (PMID: 34058980) needs to be worked out much more clearly. It remains missing in the manuscript as a figure with appropriate space in the revised text. The transcript boundaries in RTD3 and Araport11 tend to be inaccurate, this is community knowledge and in case the authors were unaware it is also clear from PMID: 34058980. It is a conceptual concern if the transcripts are also present in prior studies examining different conditions. The table inserted in the Reviewer response does not address my point. The exercise

for the revision had been to truly engage with the high-quality transcriptome annotation of Ivanov et al, ideally match that quality, and convince that the authors are largely adding new information, rather than concealing a rediscovery of what is already known and giving it a new name. I had the impression that there is a big overlap based on the detailed comments on loci were the authors presented genome browser views— many of them were not specific to the mutants. An overlap of 56,979/72,870 as indicated for 90% indicates that there is a large overlap between high confidence transcripts. The task the authors were given in the review process was to thoroughly address that systematically.

Our response: We already addressed this comment in the first round of revisions by amending the text and citing additional publications. We found these additional comments were not based on facts and might also stem from misunderstandings of the experimental approach of our study, which we further clarify below. In addition, we have performed additional investigations to illustrate the higher quality, resolution, and diversity of DRS-AtRTD3 transcriptome dataset in this study compared to the transcriptome of the study “Ivanov et al. 2021” (PMID: 34058980) that we found has serious flaws about the accuracy of transcript boundaries.

First of all, the study “Ivanov et al. 2021” (PMID: 34058980) mentioned by the Reviewer was published in May 2021 and did not evaluate the transcript boundaries of the novel transcriptome of *Arabidopsis thaliana* Reference Transcript Dataset 3 (AtRTD3), which our study used. AtRTD3 was first available online on September 2021 on bioRxiv (<https://doi.org/10.1101/2021.09.02.458763>) and published on July 2022 in Genome Biology (<https://doi.org/10.1186/s13059-022-02711-0>). The AtRTD3 transcriptome, which was released later than “Ivanov et al. 2021”, has been developed with three major updates from AtRTD1 to AtRTD3 for the improvement by continuous efforts by a consortium consisting of many laboratories from different countries. The datasets have been widely accepted as valuable reference transcriptome resources in the research community. AtRTD3 is based on experimental data validations and consists of transcripts supported by high-confidence TSS/TES data. In the paper of AtRTD3 (Zhang et al., 2022), they described as *”We developed (1) a splice junction-centric approach that allows the identification of high-confidence SJs and (2) a probabilistic 5’ and 3’ end determination method that effectively removes transcript fragments and identifies dominant transcript start and end sites. They allow accurate determination of SJs, TSS, and TES directly from the Iso-seq data and remove the requirement for hybrid error correction or parallel experimental approaches for detecting TSS and TES such as CAGE-seq or poly(A)-seq, respectively. The defined sets of high-confidence SJs, TSS, and TES were used to generate an Iso-seq-based*

transcriptome (AtIso) consisting of transcripts with accurately defined 5' and 3' ends and SJIs and the combination of AS events with specific TSS and TES.”.

Thus, the AtRTD3 was built with awareness and high confidence regarding the boundaries of transcripts, and it was misleading by the Reviewer to depreciate the quality of AtRTD3 without any analysis.

Nonetheless, here we would like to emphasize that the major goals and outcomes of this study are neither the building of a new reference transcriptome nor the evaluation of previous datasets, which the Reviewer might have misunderstood. The AtRTD3 dataset already includes RNA transcripts “*extracted from nineteen samples from different Arabidopsis Col-0 organs, developmental stages, abiotic stress conditions, infection with different pathogens and RNA degradation mutants to capture a broad diversity of transcripts*” (Zhang et al., 2022), which is suitable for our study for comprehensive detection of chimeric TE-gene transcripts present in the Arabidopsis transcriptome. Thus, the comment by the Reviewer “*It is a conceptual concern if the transcripts are also present in prior studies examining different conditions.*” is an irrelevant concern to this study, since transcripts detected in various development stages and conditions are already included in the DRS-AtRTD3 as a priori dataset. On the other hand, the transcriptome dataset in “Ivanov et al. 2021” (PMID: 34058980) used the data generated from the previous publications by Parker et al., 2020 (PMID: 31931956, also used in our DRS-AtRTD3 transcriptome) and nascent RNA-seq dataset (Kindgren et al. 2019), which were based on seedling of Arabidopsis Col-0 (10 days and 2-week-old seedlings), representing a limited set of transcriptome diversity in Arabidopsis.

In addition, the comment “*I had the impression that there is a big overlap based on the detailed comments on loci were the authors presented genome browser views— many of them were not specific to the mutants.*”, is also based on a misunderstanding of our experimental methods, as the DRS-AtRTD3 does not contain DRS data of epigenome mutants. The epigenome mutant-specific TE-gene transcripts and statistically significant isoform switching events detection were performed with a distinct strategy (Sup. Fig. 4) utilizing Illumina short reads, and epigenome data, including methylome and chip-seq data (Fig. 3, Sup. Fig. 7-10).

Even though we think a comparison of our DRS-AtRTD3 transcriptome with previous studies is irrelevant to the scope of this study as elucidated above, we further tried to systematically assess it with the data by “Ivanov et al. 2021” (PMID: 34058980) as suggested by the Reviewer. However, we found serious misannotation in the published BED12 format file (available in the GitHub of the author), in which exon-intron

boundaries were not correctly annotated (attached Figures A and B), and did not match to the canonical GT – AG splicing donor-acceptor site motifs (see for instance this study: <https://doi.org/10.1093/pcp/pct001>). The misannotation in the entire transcriptome made us difficult to conduct a direct comparison of the exon-intron structure of transcripts (attached Figure C). Based on the misannotation at the exon-intron boundaries, which had not been updated/corrected for 2 years since the publication, we cannot agree to the claim put forth by Reviewer#2 that the transcriptome from Ivanov et al. 2021 (PMID: 34058980) is a “high-quality transcriptome annotation”. Still, we tried to compare the transcriptomes (72,870 transcripts) with our DRS-AtRTD3 transcriptome (199,489 transcripts) dataset at base level by GffCompare (Pertea et al. 2020) with the default setting. As indicated in the figure below (attached Figure C), we detected that over 97 % of transcripts of Ivanov et al. 2021 were covered by DRS-AtRTD3 (this can be higher, as DRS-AtRTD3 does not include transcripts of Chloroplast and Mitochondria), that is about 51.8 % of our DRS-AtRTD3 at base level. This is not surprising to us, since DRS-AtRTD3 includes their original dataset of Parker et al., 2020 as a part of the dataset. As an effort to show additional evidence of the accuracy of the DRS-AtRTD3 transcriptome mainly used in our study, we also performed a systematic comparison of the DRS-AtRTD3 (this study) and the original AtRTD3 (Zhang et al. 2022) transcriptomes. The comparison demonstrates that DRS-AtRTD3 contains at base level 100% of the transcripts of the original AtRTD3 (Zhang et al. 2022) (attached Figure D). DRS-AtRTD3 contains additional TE-genes transcript (Fig. 1 and Table 4) and has high sensitivity and precision at exonic and intronic levels (attached Figure D). This data in the Figure D was added in the revised manuscript as Sup. Table 3 for the transparency of our data analysis.

In conclusion, we provided additional evidence that DRS-AtRTD3 adequately captures the diversity of Arabidopsis transcriptome in various developmental stages and conditions suitable for the scope of this study.

Example with AT1G01010

Figure A: Screenshot of *A. thaliana* annotations and alignments of nanopore DRS data (Col-0) in IGV (Integrative Genome Viewer) at *AT1G01010* locus. The borders at exon-intron (splicing sites) of *AT1G01010* are consistent in AtRTD3 (Zhang et al. 2022) and DRS-AtRTD3 (our study), as highlighted by the alignment of the nanopore DRS data and the canonical GT – AG splicing donor-acceptor site motifs. In contrast, exon-intron borders are inaccurate in the transcriptome “Called_transcripts.bed” from the paper of Ivanov et al. 2021 (PMID: 34058980). This error appears for all spliced transcripts as shown in Figure B. We used original files of DRS-AtRTD3 (GTF format), atRTD3 (BED12 format) and “Called_transcripts” (BED12 format) for the analysis.

Figure B: Nucleotide enrichments at exon-intron borders (splicing sites) in transcripts of DRS-AtRTD3 (our study), AtRTD3 (Zhang et al. 2022) and the “Called_transcripts.bed” (from Ivanov et al. 2021). The canonical GT – AG splicing sites are enriched in exon-intron borders in AtRTD3 and DRS-AtRTD3 transcriptome datasets, while inaccurate for the transcript annotations of Ivanov et al. 2021. Although only splicing sites of forwards transcripts (+ strand) (Chr1 to Chr5) are shown in this figure, the same issue appears in reverse transcripts (- strands, not shown).

A systematic comparison of Called_transcripts (PMID:34058980) and DRS-AtRTD3 (this study)

```

1) gffcompare -r DRS-AtRTD3.sorted.gtf Called_transcripts.gtf
#
#=# Summary for dataset: Called_transcripts.gtf
#   Query mRNAs : 72870 in 21055 loci (63697 multi-exon transcripts)
#   (12310 multi-transcript loci, ~3.5 transcripts per locus)
# Reference mRNAs : 183406 in 39744 loci (160743 multi-exon) *(15785/199489 were removed as duplicate by GffCompare)
# Super-loci w/ reference transcripts: 19562
#-----| Sensitivity | Precision |
#   Base level: 51.8 | 97.6 |
#   Exon level: 0.9 | 2.2 |
#   Intron level: 0.0 | 0.0 |
# Intron chain level: 0.0 | 0.0 |
#   Transcript level: 2.4 | 6.0 |
#   Locus level: 10.7 | 20.5 |

2) #gffcompare DRS-AtRTD3.sorted.gtf -r Called_transcripts.gtf
#
#=# Summary for dataset: DRS-AtRTD3.sorted.gtf
#   Query mRNAs : 199489 in 40042 loci (174656 multi-exon transcripts)
#   (19018 multi-transcript loci, ~5.0 transcripts per locus)
# Reference mRNAs : 58538 in 21055 loci (51313 multi-exon) *(14332/72870 were removed as duplicate by GffCompare)
# Super-loci w/ reference transcripts: 19562
#-----| Sensitivity | Precision |
#   Base level: 97.6 | 51.8 |
#   Exon level: 2.0 | 0.9 |
#   Intron level: 0.0 | 0.0 |
# Intron chain level: 0.0 | 0.0 |
#   Transcript level: 8.1 | 2.4 |
#   Locus level: 22.4 | 11.6 |

```

Figure C: A systematic comparison between the transcriptome “Called_transcripts.bed” (Ivanov et al. 2021) and DRS-AtRTD3 (transcriptome in this study) using the tool GffCompare (<https://doi.org/10.12688/f1000research.23297.1>) The direct comparisons was made with 1) DRS-AtRTD3 or 2) “Called_transcripts.bed” as a reference. The original “Called_transcripts.bed” (0-based coordinates) was converted into the GTF file format (with the 1-based coordinates) as input for GffCompare. The sensitivity and precision at base level is highlighted in red. Metrics are defined in the GFFCompare manual. The low percentage of sensitivity and precision for the comparison of exon and intron sequences is caused by the inaccuracy of exon-intron borders in the (Ivanov et al. 2021) transcriptome (as shown in Figures A and B).

A systematic comparison of AtRTD3 (Zhang et al. 2022) and DRS-AtRTD3 (this study)

```
1) gffcompare -r AtRTD3.gtf DRS-AtRTD3.sorted.gtf
#
#=# Summary for dataset: DRS-AtRTD3.sorted.gtf
#   Query mRNAs : 199489 in 40042 loci (174656 multi-exon transcripts)
#   (19018 multi-transcript loci, ~5.0 transcripts per locus)
#   Reference mRNAs : 158314 in 37959 loci (137151 multi-exon) *(10635/168949 were removed as duplicate by GffCompare)
#   Super-loci w/ reference transcripts: 37242
#-----| Sensitivity | Precision |
#   Base level: 100.0 | 91.0 |
#   Exon level: 92.4 | 89.1 |
#   Intron level: 100.0 | 98.0 |
#   Intron chain level: 98.8 | 77.6 |
#   Transcript level: 97.7 | 77.5 |
#   Locus level: 98.0 | 91.2 |

2) #gffcompare AtRTD3.gtf -r DRS-AtRTD3.sorted.gtf
#
#=# Summary for dataset: AtRTD3.gtf
#   Query mRNAs : 168949 in 37959 loci (146535 multi-exon transcripts)
#   (18810 multi-transcript loci, ~4.5 transcripts per locus)
#   Reference mRNAs : 183552 in 39744 loci (160871 multi-exon) *(15639/199489 were removed as duplicate by GffCompare)
#   Super-loci w/ reference transcripts: 37242
#-----| Sensitivity | Precision |
#   Base level: 91.1 | 100.0 |
#   Exon level: 89.3 | 91.6 |
#   Intron level: 98.1 | 100.0 |
#   Intron chain level: 80.9 | 88.8 |
#   Transcript level: 80.7 | 87.6 |
#   Locus level: 91.2 | 97.2 |
```

Figure D: A systematic comparison between the transcriptomes AtRTD3 (Zhang et al. 2022) and DRS-AtRTD3 (transcriptome in this study) using GffCompare (<https://doi.org/10.12688/f1000research.23297.1>) The direct comparisons was made with 1) AtRTD3 or 2) DRS-AtRTD3 as a reference. The sensitivity and precision at base level were highlighted in red. Metrics are defined in the GFFCompare manual. Chloroplast and Mitochondria transcripts of AtRTD3 were removed because they are not included in DRS-AtRTD3.

This comparison demonstrates that DRS-AtRTD3, which made it possible to resolve new TE-G transcripts (See revised manuscript, Figure 1a and Sup. Table 3), has high sensitivity and precision.

- I remain unconvinced about how the repeats of the ONT data were used. While repeats were performed, they were merged without harnessing the benefits of generating repeats. For example by implementing criteria to call novel transcripts only if present in several repeats etc to reach a quality threshold. This strategy deprives the authors of the opportunity to carefully perform QC on the number of reads from independent experiments that support novel transcripts.

Our responses: Since Nanopore DRS sequencing yields a relatively low read amount for each biological replicate, QC process using separate replicates has not been practically conducted in previous DRS studies such as Parker et al.

2021 (<https://doi.org/10.7554/eLife.65537>), Parker et al. 2022

(<https://doi.org/10.7554/eLife.78808>) and the “Invanov et al. 2021” (PMID: 34058980) the Reviewer mentioned. As we already commented to the Reviewer in the first revision, we

generated de novo DRS transcriptome of Col-0 with Stringtie2 and merged to the reference transcriptome AtRTD3. The same method was applied by others in Parker et al. 2021 (<https://doi.org/10.7554/eLife.65537>) and Parker et al. 2022 (<https://doi.org/10.7554/eLife.78808>).

Using a prior ONT DRS set is somewhat OK, but it is of course much better to generate their own. How can the authors confidently rule out that a portion of the transcript isoform differences between the mutants and the highly merged Col dataset that is largely contributing to the control set is actually rooted in differences in growth conditions?

Our responses: The comments suggest that the Reviewer might have missed the experimental dataset and approaches described in the manuscript. We performed our own DRS for Col-0, as well as for epigenome mutants (See Supplementary Data 1 and Sup. Table 1) at the identical developmental stage. Differential expression of isoforms and isoform switching events were further verified by our own Illumina short read RNAseq data and epigenome data for detection of their significant changes (Fig. 3, Sup. Fig.4, Sup. Fig. 7-10). We confirmed the validity of our methodology by examining the presence/absence of known epigenetically-regulated transcripts in our results (such as *SQNAT2G15790* in Fig. 3b and *RPP4* locus in Fig. 4a). We have also experimentally validated the epigenetic regulation of many candidate transcripts at selected loci by PCR and qPCR (Fig. 5b, c; Fig. 6; Sup. Fig. 14b, c, d; Sup. Fig. 15b; Sup. Fig. 18b; Sup. Fig. 19b; Sup. Fig. 23a).

Remaining issues:

- The authors claim that ONT DRS is a new method. That was true a few years ago, but by now it is already around for many years. Issues such as the abortion of reads, resulting in the impression of artifacts misleadingly indicating 5'-isoforms are known issues. Next-level analyses are already published, for example done in PMID: 34058980. Simply using ONT DRS without addressing these biases is actually a step backwards from the knowledge foundation in 2022/23 rather than an advance.

Our response: We have not used the word “new” for ONT DRS technology in the original or the revised manuscript. We are aware of the issue about 5'-truncated DRS reads leading to a potential bias of ONT-DRS coverage towards 3' ends of transcripts, which was also pointed out by the Reviewer#1. To address the issue, we have included ONT-DRS data from Parker et al 2020, obtained by cap-dependent ligation of a biotinylated 5' adapter RNA to enrich capped mRNAs. In addition, we merged it to the latest reference transcriptome AtRTD3 (Zhang et al., 2022) released later than Ivanov et al. 2021 (PMID:

34058980), which consists of transcripts supported by high-confidence TSS/TES data as quoted in the response above, to obtain a non-redundant set of transcript annotation. The DRS-AtRTD3 dataset covered >97% of PMID: 34058980 at basal level as analyzed above. Thus, the DRS data from capped transcripts and AtRTD3 would have minimized the effects of truncated transcripts compared to DRS-only transcriptome analysis. However, here we would like to emphasize again that this study's major goals and outcomes are neither the building of a new reference transcriptome nor the evaluation of, or the comparison with previously published datasets.

- The authors worked on the readability regarding the previous acronyms to some extent. The benefit of using the author-defined acronyms over community standards remains unclear. Chimeric transcript have been studied and so there is no real need to reinvent the wheel.

Our response: In this manuscript, we did not invent new words, and the term “chimeric TE-gene transcript” has been widely used in previous studies, even in other organisms such as human (Lock et al. 2017; <https://doi.org/10.1371/journal.pone.0180659>). Other acronyms are abbreviations of combined common words in the biology field, which are introduced for the readability of the text by following the suggestion by the Reviewer#3. Acronyms were defined at first mention in the text as a rule for the manuscript writing.

- The information on propagation of chromatin mutants remains missing from the methods.

Our response: As suggested, we added the strain information in the method of the revised manuscript.

- The authors should clarify the ‘-R’ option of in the in the StringTie2 manual. A comparison of -L and -R modes and a valid, scientific comparison of both modes should be presented.

Our response: As suggested, we added the “-R” option information according to the manual of StringTie2 in the revised manuscript. A scientific comparison of outputs using -L or -R modes for both Araport11 and AtRTD3 as references was already presented in Sup. Fig. 1b.

Reviewer #3 (Remarks to the Author):

The Authors addressed carefully my comments and I found just two remaining minor points to consider.

1. legends for the supplementary figures appear to be missing.

Our response: We have added figure legends directly under the supplementary figures.

2. *“As commented, the 39,998 gene models of DRS-AtRTD3 include transcripts in Chr1 to 5 of AtRTD3, therefore it also contains lncRNA genes, ribosomal RNA genes, transfer-RNA genes, micro-RNA genes in addition to the protein-coding genes. We provided this information in the method section.”*

I find it curious to represent the percentage of gene isoforms relative to the total number of genes (fig 1d) with the inclusion of lncRNA genes, ribosomal RNA genes, transfer-RNA genes, micro-RNA genes which are not spliced. This tends to exaggerate the perception that the phenomena described by the authors in this paper is only marginal. Could the Authors redo the calculation with only RNAs that may be prone to splicing?

Our response: As suggested, we reanalyzed DRS-AtRTD3 and found transcripts overlapping (> 80%) with lncRNA genes, ribosomal RNA genes, tRNA genes and miRNA genes, corresponding to 4,395/39,998 gene models. These include 68/1,171 genes with ATE-G isoforms. Thus, if these genes are excluded, genes with ATE-G isoforms become 1,103/35,603 (3.1%) of total genes without the exclusion, which is largely unchanged from the original calculation. Actually, miRNA gene transcript can be alternatively spliced (<https://doi.org/10.1002/wrna.1403>), and about 50% of annotated Arabidopsis lncRNA genes are reported to contain introns (<https://doi.org/10.1016/j.pbi.2015.02.008>). Thus, we think those gene annotations are relevant to the study and should be included in the analysis.